



# Intercomparison of MAX-DOAS, FTIR and direct sun HCHO vertical columns at Xianghe, China

Gaia Pinardi[1], Martina M. Friedrich[1], Corinne Vigouroux[1], Bavo Langerock[1], Isabelle De Smedt[1], Caroline Fayt[1], Christian Hermans[1], Steffen Beirle[2], Thomas Wagner[2], Minqiang Zhou[1, 3,4], Ting Wang[3], Pucai Wang[3], Martine De Mazière[1], and Michel Van Roozendael[1]

[1]Royal Belgian Institute for Space Aeronomy (BIRA-IASB), Av Circulaire 3, 1180 Uccle, Belgium
[2]Max-Planck-Institut für Chemie (MPIC), Hahn-Meitner-Weg 1, 55128 Mainz, Germany
[3]Institute of Atmospheric Physics (IAP), Chinese Academy of Sciences, Beijing, China
[4]State Key Laboratory of Atmospheric Environment end Extreme Meteorology, Institute of Atmospheric Physics, Chinese Academy of Sciences, Beijing 100029, China

**Correspondence:** Gaia Pinardi (gaia.pinardi@aeronomie.be)

**Abstract.** MAX-DOAS (Multi-AXis Differential Optical Absorption Spectroscopy), direct sun DOAS (DS) and FTIR (Fourier Transform InfraRed) measurements are considered nowadays as reference data for the validation of HCHO satellite observations. Recognizing their strengths and limitations, as well as evaluating their consistency, is crucial for generating robust and reliable validation datasets. So far, only a handful of studies have explored the complementarity between MAX-DOAS and direct sun FTIR HCHO measurements. Here we take advantage of the presence of a MAX-DOAS spectrometer, incorporating a direct sun viewing mode capability, and an FTIR instrument operating in parallel at the Xianghe site (39.75°N, 116.96°E, China), to compare the retrieved HCHO vertical columns and investigate in detail the reasons for the observed differences. First, we compare the UV and IR HCHO vertical columns in the direct sun geometry, for which the uncertainty due to the light path is negligible. We find an excellent agreement between the measurements obtained in both wavelength ranges, with a median difference of less than -0.5 $x10^{15}$ molec/cm$^2$ (-6%). Second, the MAX-DOAS data are compared to the DS and FTIR ones. The study addresses the impact of using different MAX-DOAS retrieval methods all implemented within the FRM4DOAS centralized processing facility, and discusses differences related to the vertical sensitivity of the measurements in each geometry.

The MAX-DOAS HCHO columns correlate well with the direct sun DOAS and FTIR data, but have a tendency to underestimate them by -22% and -20.8% respectively. This bias can be reduced to 1% when taking properly into account the different a-priori profiles and the respective vertical sensitivities of the MAX-DOAS and FTIR measurements. If the comparison is restricted to the 0-4km altitude range where MAX-DOAS measurements have their best sensitivity, differences are further reduced with a bias of about -4.6% for the original comparison and of 2.5% after taking into account the respective vertical sensitivities and a priori profiles. The underestimation in the retrieved MAX-DOAS vertical columns (VCDs) is shown to be due to the choice of the a priori profile, which neglects the free-tropospheric contribution (above 4km), where the MAX-DOAS has no sensitivity. These results suggest that improvements to the current FRM4DOAS MAX-DOAS HCHO retrievals are possible.



We investigate whether the underestimation of the MAX-DOAS tropospheric VCDs can be reduced by using more appropriate a priori profiles, based on the CAMS and TM5 chemical-transport models (CTMs). We illustrate the bias reduction with respect to our reference direct sun data. The impact is different depending on the season. The use of CTM-based a priori profiles has a positive impact on the retrieved VCDs in all seasons except winter. When restricting the comparison to the 0-4km altitude range, the impact of the a priori profile is only significant in the winter period, also leading to a degradation of the agreement with FTIR data. The improvement of the agreement between MAX-DOAS and FTIR data is thus mainly related to a better handling of the free-tropospheric part of the profile, smaller in winter than in other seasons.

# 1  Introduction

Formaldehyde (HCHO), the most abundant carbonyl compound in the atmosphere, is an important atmospheric trace gas which is produced by the photochemical degradation of methane as well as non-methane volatile organic compounds (NMVOCs), the latter being the main source of HCHO over continents (Finlayson-Pitts and Pitts, 2000; Wolfe et al., 2016). It is in particular a byproduct of the oxidation of isoprene (Stavrakou et al., 2009). The primary emission sources of HCHO are biomass burning, industrial processes, fossil fuel combustion and vegetation (see e.g., Wolfe et al., 2016, and references therein). HCHO is frequently used as a proxy for NMVOCs, which are crucial in the photochemical processes that influence air quality. It is a precursor of the formation of ozone and aerosols, and it participates to photochemical smog production, especially in urban areas (Atkinson, 2000). HCHO is destroyed by photolysis and reaction with OH. Characterized by a short atmospheric lifetime (ranging from hours to a day), HCHO is highly reactive and its abundance reflects recent VOC oxidation. This molecule plays a pivotal role in the fast-paced dynamics of air quality, making it an important species to monitor in real-time pollution assessments.

HCHO can be measured in both the UV and IR part of the solar spectrum, from ground-based and satellite platforms.

The Multi-Axis Differential Optical Absorption Spectroscopy (MAX-DOAS, Honninger et al., 2004; Wagner et al., 2004; Frieß et al., 2006) is a passive remote sensing technique that measures scattered sunlight at different elevation angles from the horizon to the zenith. It is sensitive to absorbers present close to the surface. From measured spectra, differential slant columns densities (DSCD) of target gases are derived by least-squares fitting to reference absorption cross-sections (Platt and Stutz, 2008) and, from there, the vertical column density (VCD) and the vertical distribution of the trace gas concentration can be retrieved by applying different inversion methods, such as the Optimal Estimation (OEM; Rodgers (2000), e.g., Hendrick et al., 2004; Honninger et al., 2004; Wittrock et al., 2004; Frieß et al., 2006; Clémer et al., 2010), parametrized approaches (Wagner et al., 2011; Vlemmix et al., 2015; Beirle et al., 2019; Irie et al., 2008) or simple geometrical approximations. MAX-DOAS HCHO measurements have been reported from many locations worldwide (e.g. Heckel et al., 2005; Wagner et al., 2011; Peters et al., 2012; Vlemmix et al., 2015; Gratsea et al., 2016; Wang et al., 2017, 2019; Hoque et al., 2018a, b; Irie et al., 2019; Benavent et al., 2019; Kumar et al., 2020; Chan et al., 2020; Schreier et al., 2020; Yombo Phaka et al., 2023; Rawat et al., 2024; Chong et al., 2024) and during intercomparison campaigns (e.g., Pinardi et al., 2013; Kreher et al., 2020; Piters et al.,



2012; Tirpitz et al., 2021). Only few studies focused on comparing results from different retrieval approaches (Vlemmix et al., 2015; Tirpitz et al., 2021; Frieß et al., 2019).

Formaldehyde has also been measured by FTIR (Fourier Transform Infrared) ground-based spectrometers, originally at a few individual stations as part of the Network for the Detection of Atmospheric Composition Change (NDACC), using different retrieval settings (Mahieu et al., 1997; Notholt et al., 1997; Hak et al., 2005; Vigouroux et al., 2009; Jones et al., 2009; Paton-

Walsh et al., 2010; Viatte et al., 2014). In Vigouroux et al. (2018), the retrieval settings have been harmonized within the whole network and continuous HCHO measurements are since then performed at more that twenty stations in a consistent way and archived in the NDACC database (https://ndacc.larc.nasa.gov/).

Another network growing fast in an harmonized way is the Pandonia Global Network (PGN, https://www.pandonia-global-network.org). PGN provides HCHO measurements using Pandora spectrometers operated both in direct sun and in multi-axis

geometries (Spinei et al., 2018, 2021; Park et al., 2018; Herman et al., 2018).

MAX-DOAS, direct sun and FTIR networks are rapidly expanding and are increasingly regarded as reference data sources for validating satellite HCHO observations, e.g. for Low Earth orbits (LEO) (Vigouroux et al., 2020; De Smedt et al., 2015, 2021; Lee et al., 2015; Yombo Phaka et al., 2023; Chan et al., 2020, 2022; Müller et al., 2024; Herman and Mao, 2024), geostationary orbit satellites (GEO) like GEMS (Lee et al., 2024; Fu et al., 2025; Bae et al., 2025) and for chemistry-transport

models (Oomen et al., 2024).

This has been made possible owing to recent improvements in retrieval techniques and harmonization of the methodologies applied at the network level, in particular for FTIR instruments (Vigouroux et al., 2018) and with the rapid development of the PGN network. For MAX-DOAS, standardized settings for the SCD retrievals have been proposed during intercomparison campaigns (Pinardi et al., 2013; Kreher et al., 2020) and the different HCHO inversion approaches lead to consistent tropo-

spheric vertical columns but to larger differences in terms of profiles (Vlemmix et al., 2015; Frieß et al., 2019; Tirpitz et al., 2021).

The lack of harmonization of the current MAX-DOAS HCHO datasets is being addressed by the recent development of a centralized processing facility for MAX-DOAS, as part of the FRM4DOAS project (Fiducial Reference Measurements for Ground-Based DOAS Air-Quality Observations, https://frm4doas.aeronomie.be/, Van Roozendael et al., 2024). FRM4DOAS

is an international project funded by the European Space Agency (ESA) aiming at harmonizing and standardizing the data retrieval of MAX-DOAS instruments operated within NDACC. HCHO columns and profiles are currently under scientific testing and require further consolidation, so they are not distributed operationally yet. They have been used in a few scientific studies (Karagkiozidis et al., 2022; Yombo Phaka et al., 2023; Oomen et al., 2024; Lange et al., 2024; Bae et al., 2025), showing very promising results and useful insight in satellite validation during campaigns and new sites development.

To produce robust and consolidated validation results, it is essential to understand the strengths and limitations of each correlative technique and to assess their consistency. In the past, only a few studies have explored the complementarity between MAX-DOAS and FTIR HCHO measurements. These were conducted in remote mountain-top observatories (Vigouroux et al., 2009; Franco et al., 2015), in remote regions (Ryan et al., 2020) and one in largely polluted area (Rivera Cárdenas et al., 2021)



around Mexico City. They led to mixed findings, with biases close to zero or up to about 30% differences depending on the

location, the HCHO abundances and its horizontal inhomogeneity.

In the present study, we take benefit of MAX-DOAS and FTIR spectrometers having been operated in parallel at the Xianghe station (China) during several years, to investigate in detail the agreement of the HCHO retrievals from these instruments in highly variable conditions. Moreover, the MAX-DOAS instrument also incorporates a direct sun viewing mode capability, allowing the comparison of direct sun measurements in the UV and in the IR.

The different instruments and retrievals are presented in Sect. 2 and their comparison is discussed in Sect. 3. In addition to comparing direct-sun DOAS and FTIR data, we investigate several MAX-DOAS retrieval approaches and compare the resulting HCHO columns to the direct sun ones. The study also addresses the impact of the different vertical sensitivities of the MAX-DOAS and FTIR techniques (Sect. 3.2.1). Overall the aims are to: 1) assess the quality of the MAX-DOAS HCHO products currently delivered by the FRM4DOAS system and 2) revisit the HCHO retrieval approach used in the system to

further improve its accuracy (Sect. 3.2.2). This work intends to benefit to the whole DOAS community, which will result in more coherent correlative datasets for satellite validation.

Compared to previous similar studies, this work compares state-of-the art harmonized retrievals strategies for both instrument types (UV-visible and FTIR) and also includes direct sun DOAS measurements. This addition is interesting considering the fast growing number of direct sun DOAS Pandora HCHO measurements within the PGN network.

**2  Instruments and Datasets**

The Xianghe Observatory (39.75° N, 116.96° E) is a suburban site close to Beijing, China operated/owned by the Institute of Atmospheric Physics (IAP), of the Chinese Academy of Sciences. A MAX-DOAS from BIRA-IASB/IAP was operated on the roof of the LAGEO laboratory (http://lageo.iap.ac.cn/) between 2010 and 2022, and a Bruker IFS 125HR FTIR spectrometer from BIRA-IASB/IAP is measuring since June 2018. The MAX-DOAS instrument also includes a direct sun channel. The

instruments and their HCHO inversions are described below.

**2.1  MAX-DOAS**

The MAX-DOAS instrument at Xianghe has been extensively described in past studies (Clémer et al., 2010; Vlemmix et al., 2015; Hendrick et al., 2014). The system has been designed and assembled at BIRA-IASB in Brussels in 2007, then operated in Beijing during the 2008 Olympic Games before participating to the CINDI-1 campaign (Piters et al., 2012; Pinardi et al., 2013)

in 2009 and finally being operated from March 2010 to mid 2022 in Xianghe. It is a dual channel system composed of two grating spectrometers covering the UV and visible wavelength ranges (300–390 nm and 400–720 nm, with a spectral resolution of 0.4 nm and 0.9 nm, respectively), connected to two cooled (-50°C) CCD detectors. The spectrometers and detectors are enclosed in a thermo-regulated box that is connected to an external part, the optical head, through optical fibers. The optical head is mounted on a commercial sun tracker (INTRA, Brusag) that allows collecting the scattered light from a series of

user-defined elevation and azimuth angles. The measurement routines includes zenith measurements at twilight and off-axis



measurements (1°, 2°, 3°, 4°, 6°, 8°, 10°, 12°, 15°, 30° elevation angles and zenith) from 85° SZA (solar zenith angle) sunrise to 85° SZA sunset with a time resolution of ~15 min. After each off-axis scan, a measurement pointing to the sun (direct sun mode, DS) is also performed by means of a diffuser plate mounted on a filter wheel. From 2010 to mid 2018 the instrument was oriented towards the south-east (165° azimuth) while from November 2019 onwards, the pointing direction was changed

to close to the North (179° azimuth wrt North) due to an obstruction in the former viewing direction.

The instrumental set up including data transfer is fully automated, allowing continuous daily operation throughout the year. Several products have been retrieved such as aerosols (Clémer et al., 2010), $NO_2$ (Hendrick et al., 2014; Vlemmix et al., 2015), HCHO (Vlemmix et al., 2015), HONO (Hendrick et al., 2014), $SO_2$ (Wang et al., 2014), CHOCHO, and have been used as reference in several satellite validation studies for LEO (e.g., De Smedt et al., 2015, 2021; Theys et al., 2015, 2021; Pinardi

et al., 2020; Compernolle et al., 2020; Verhoelst et al., 2021; Lerot et al., 2021) and GEO sensors (Lee et al., 2024; Ha et al., 2024).

The Xianghe MAX-DOAS measurements cover the period from March 2010 to July 2018 and from October 2019 to November 2021 for the UV channel. The VIS channel continued to work until August 2022. In this study, we concentrate on the HCHO data product developed within the FRM4DOAS project, and we focus on the period from 2018 onwards, when both the

FRM4DOAS system (started in July 2018, see Van Roozendael et al. (2024)) and the FTIR instrument were in operation, see Fig. 1.

The measured spectra are analyzed with the QDOAS software (http://uv-vis.aeronomie.be/software/QDOAS/, Danckaert et al. (2017)), providing DSCDs that correspond to the concentration of the absorber integrated along the effective light path of the recorded sunlight, relative to the amount in the measured reference spectrum. The QDOAS settings used are fully described

in the FRM4DOAS Algorithm Theoretical Basis Document (ATBD) (Hendrick et al., 2018), and are summarized in Table 1.

As mentioned in Sect. 1, different types of algorithm can be used to retrieve information about the vertical distribution of aerosols and trace gases from slant columns retrieved within a sequence of elevation angles (the off-axis sequence). Below we describe the specificities of the FRM4DOAS centralized processing system (Van Roozendael et al., 2024), that have been used here for the MAX-DOAS HCHO profile retrievals.

FRM4DOAS incorporates community-based retrieval algorithms into a fully traceable, automated and quality controlled processing environment that provides harmonized vertical columns and profiles for satellite validation. Two codes have been selected for implementation: 1) the MMF (Friedrich et al., 2019) optimal estimation-based algorithm and 2) the MAPA (Beirle et al., 2019) algorithm, which is based on a parametrization of the retrieval profile shape and a Monte-Carlo approach for the inversion, see details below.

Both MMF and MAPA codes implement a two-step retrieval approach for trace gas profile retrieval. In the first step, the aerosol profile is determined based on a set of $O_4$ DSCDs. The $O_4$ vertical profile is well-known (it varies with the square of the $O_2$ monomer concentration), and $O_4$ DSCD measurements are used to provide information on the aerosols optical depth (AOD) and the vertical distribution of aerosols (Wagner et al., 2004; Frieß et al., 2006). In the second step, the retrieved aerosol profile is used to constrain the radiative transfer simulations needed for the trace gas retrieval. Both codes start from the same





**Figure 1.** HCHO VCD time series derived in Xianghe (China) from FTIR, direct-sun DOAS, and two instances of MAXDOAS retrievals (MMF, and MAPA).

$O_4$ and HCHO DSCD and share the same meteorological input parameters, while other parameters are algorithm specifics, as summarized in Table 2.

The current strategy in FRM4DOAS is to run both MMF and MAPA inversions independently, so to produce aerosols, $NO_2$ and HCHO tropospheric profiles and columns from both algorithms in near real-time (NRT). For $NO_2$, there is an additional merging step, that is checking the coherence of both inversion results in order to provide consolidated





**Table 1.** DOAS analysis settings.

| Instrument | wavelength range | cross-sections | other |
|---|---|---|---|
| MAX-DOAS | 324.5-359 nm | HCHO (297 K): Meller and Moortgat (2000) | Polynomial degree: |
| | | $NO_2$ (298 K): Vandaele et al. (1998) | Order 5 (6 coefficients) |
| | | $O_3$ (223,243 K): Serdyuchenko et al. (2014)* | Intensity offset: |
| | | $O_4$ (293 K): Thalman and Volkamer (2013) | Order 1 |
| | | BrO (223 K):Fleischmann et al. (2004) | Calibration: based on |
| | | Ring: QDOAS high resolution based on SAO: | reference SAO |
| | | Chance and Kurucz (2010) | Chance and Kurucz (2010) |
| | | | reference of the scan (average) |
| direct sun | 324.5-359 nm | HCHO (297 K): Meller and Moortgat [2000] | Polynomial degree: |
| | | $NO_2$ (298 K): Vandaele et al. (1998) | Order 5 (6 coefficients) |
| | | $O_3$ (223, 243K K): Serdyuchenko et al. (2014)* | Intensity off-set: |
| | | $O_4$ Finkenzeller and Volkamer (2022) | Order 2 |
| | | BrO (223 K): Fleischmann et al. (2004) | Calibration: based on reference |
| | | HONO: Stutz et al. (2000) | SAO Chance and Kurucz (2010) |

* pre-shift of +0.003 nm.

datasets for operational delivery outside of the FRM4DOAS consortium, as harmonized GEOMS format files (GEOMS hdf, https://evdc.esa.int/documentation/geoms/). The integrated $NO_2$ partial column up to 2 km and up to 4 km from both algorithms are compared and retained if they agree within their combined uncertainties, and at least one of the individual codes needs to judge a scan as valid (Van Roozendael et al., 2024). The $NO_2$ data are currently in a pre-operational state, with daily transfer of the $NO_2$ GEOMS files to the NDACC rapid delivery database (RD, ftp://ftp.cpc.ncep.noaa.gov/ndacc/RD/), including a mirroring on the EVDC (ESA Validation Data Center, https://evdc.esa.int/), as described in Van Roozendael et al. (2024).

For HCHO profiles, this procedure does not work as well as for $NO_2$, due to the fact that MMF and MAPA results differ in too many cases, leading to a limited number of consolidated data. As a result, the product is still under development. In this paper we compare MMF and MAPA HCHO profile inversion results to our reference direct sun data (see Sect. 3), and we investigate options to improve the retrievals (Sect. 3.2.2).

### 2.1.1 MMF profile retrievals

The Mexican MAX-DOAS Fit (MMF, Friedrich et al. (2019)), is an inversion algorithm using constrained least-square fitting with an optimal estimation regularization. The intensity mode from VLIDORT 2.7 (Spurr et al., 2001) radiative transfer model is used as forward model and a least-square fitting approach with Levenberg-Marquard iteration as presented in Rodgers (2000) is followed. On each iteration the new state vector $x$ (the gas vertical profile) is calculated from the measurement vector $y$ (a set





of DSCDs measured at the different elevation angles forming the scan), the a priori profile $x_a$ in concentration (molec/cm$^3$), the measured error covariance matrix $S_m$, the a priori covariance matrix $S_a$, the simulated DSCD of the previous iteration $F(x_i)$ and the Jacobians of the previous iteration $K_i$, as in Eq. (A1) of Friedrich et al. (2019) (Eq. (5.36) of Rodgers, 2000).

The residuals from the QDOAS fitting are in most cases dominated by the random noise of the detector. The measurement
error covariance matrix $S_\epsilon$ is chosen diagonal with values corresponding to the statistical errors on the trace gas QDOAS fitting. This neglects the errors due to temporal changes between measurements.

In the used MMF version, the a priori profile and the associated a priori covariance matrix $S_a$ are constructed as follows. The standard a priori profile $x_a$ is constant in time, and corresponds to an exponentially decreasing (with height) profile defined by the a priori vertical column $VCD_a$ and a scaling height $SH$:

$$x_a(z) = \frac{VCD_a}{SH} exp(-z/SH) \tag{1}$$

where z is the center of the altitude grid above the surface.

The covariance matrices $S_a$ are constructed from the a priori profile, by considering a variance on the diagonal elements $S_{a\ i,i} = (SaScal * x_{ai})^2$ and using Gaussian functions to calculate off-diagonal elements $S_{a\ i,j}$, following Hendrick et al. (2004). These account for correlations between trace gas concentrations at different altitude levels :

$$\begin{aligned} S_{a\ i,j} &= \sqrt{S_{a\ i,i} S_{a\ j,j}} \exp(-ln(2)(\frac{z_i - z_j}{\eta})^2) \\ &= SaScal\ x_{ai} x_{aj} \exp(-|\frac{z_i - z_j}{\eta}|) \end{aligned} \tag{2}$$

For HCHO, $SH$ is set to 1km, $VCD_a$ is 8.4 $\times 10^{15}$ molec/cm$^2$, the constrain on the diagonal (the scale parameter $SaScal$) is set to 0.5 and the correlation length $\eta$=200 m (see Table 2).

The vertical sensitivity of a retrieved profile $x$ to the true profile $x_{true}$ is described by the averaging kernel matrix $A$:

$$A = \partial x / \partial x_{true} \tag{3}$$

$A$ is calculated by MMF for each retrieved profile as in Eq.(19) of Friedrich et al. (2019). Each row of $A$ is called an averaging kernel and is associated with the retrieved profile point at a certain altitude. The averaging kernel describes the vertical sensitivity function and how the retrieval distributes a deviation from the true profile into the different vertical layers. It is fundamentally linked to the vertical resolution of the retrieval. Specifically, the trace of $A$ provides an estimate of the number
of degrees of freedom for signal (DOFs) which corresponds to the number of independent pieces of information that can be retrieved from the measurements (Rodgers, 2000). The column averaging kernel $AK_{col}$ (the sum of the individual averaging kernels rows) represents the sensitivity of the VCD to the changes in different heights. In an ideal case, it would be around 1 at all altitudes.

An illustration of the averaging kernels for the MAX-DOAS MMF case is given in Fig. 2a for the 1rst of July 2020. It can be
seen that for the lowest layers close to the surface, the rows of $A$ have values close to 1, while the sensitivity rapidly decreases




above around 1.5km, reaching values close to zero, where all the information content is coming from the a priori used in the OE inversion. The median DOF for that day is 2.84, sign that there are at least 2 independent pieces of information to be extracted from the profiles. For the whole 2020, the median DOF is of 2.62. The a priori profiles and the cumulative partial column DOF can be seen in Fig. 10 for February and July 2020.

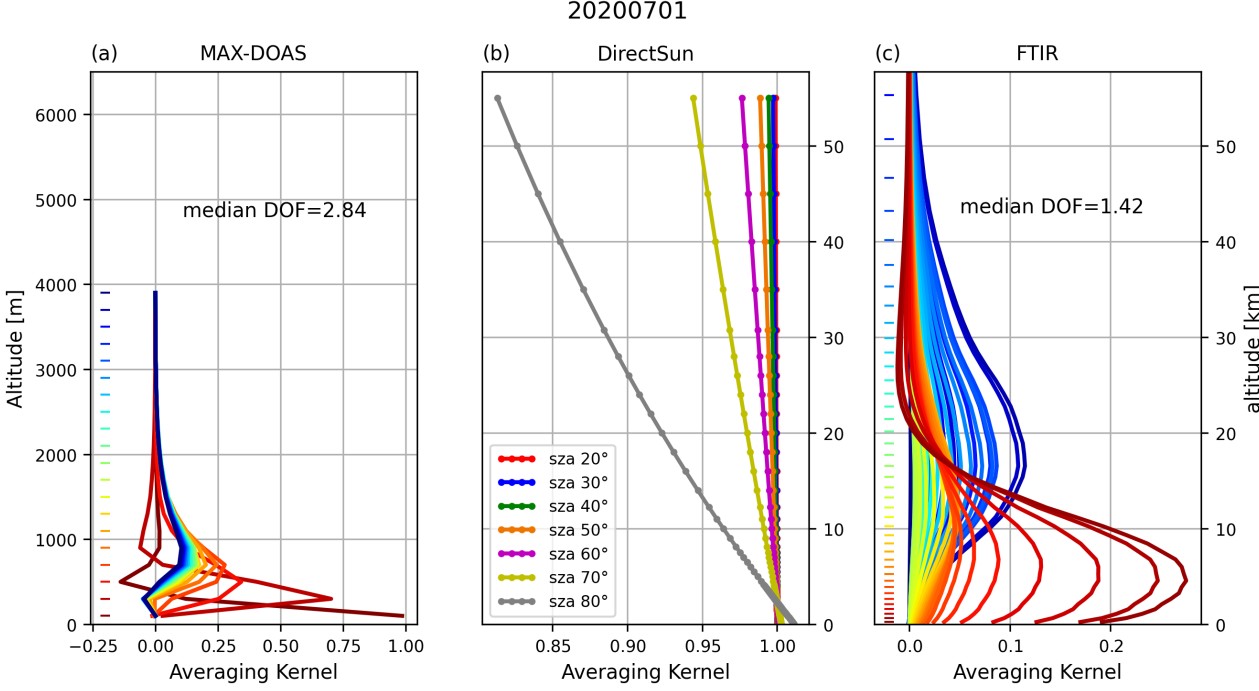

**Figure 2.** MAX-DOAS, direct sun and FTIR HCHO averaging kernels for 1/7/2020. The median DOF for the day is given as insert. For the direct sun, the typical vertical sensitivity is given as a function of the solar zenith angle (SZA). Note the different altitude scales used for MAX-DOAS (in m) and for direct sun and FTIR data (in km).

In the following, we will consider the MMF data with its own recommended filtering criteria ($qa_{mmf} < 2$, i.e. keep valid and warning cases). Three conditions should be met. First, scans with a DOF below 1 are excluded. Secondly, all scans with an average dSCD root mean square (rms) larger than 4 times the DSCD retrieval error are excluded. Finally, due to a lack of good a priori knowledge for the aerosols, two aerosol retrievals are performed (differing by a factor 10 in AOD). If the retrieved aerosol profiles agree well, only one trace gas retrieval is performed and no extra test is applied. If the retrieved aerosol profiles

differ by more than 20% (as averaged partial AOD in each layer), the trace gas profile retrieval is performed with both aerosol profiles, and all scans for which the retrieved HCHO VCD differs by more than 10% are flagged as invalid.

    The errors on the MAX-DOAS VCD are discussed in Sect. A1, along with the direct sun and the FTIR contributions. The error, as the percentage value wrt the VCD columns, is presented as the median value within 5° SZA bins from 15° to 80° SZA, see Fig. A1. MMF errors are of the order of 10 to 15% of the VCD.



### 2.1.2  MAPA profile retrievals


The Mainz Profile Algorithm (MAPA, Beirle et al. (2019)) is a profiling algorithm that is based on a parametrization approach. Atmospheric profiles are parametrized using three parameters: the integrated column c, the layer height h and the shape parameter s. The main reason to use a low number of free parameters is that the information content of MAX-DOAS observations with respect to the vertical distribution of aerosols and trace gases is limited, and therefore, a sufficiently wide range of possible
profile shapes can be retrieved with a limited but appropriate choice of free parameters. With the MAPA choices

- – s=1 corresponds to a box profile with height h, containing column c

- – For s<1, the fraction s*c is within the box of height h, and (1-s)*c above (exponentially decreasing)

- – s>1 represents elevated layers.

Additionally, a fourth optional parameter can be included, the $O_4$ scaling factor, which was initially introduced by Wagner
et al. (2009) in order to achieve agreement between the measured DSCDs and the forward model simulations. As currently there is no consensus for the need or not of such scaling factor or on its explanation (Wagner et al., 2019), three sets of MAPA outputs are run for different possibilities of the $O_4$ scaling factor within the FRM4DOAS centralized processing implementation: using a fixed 0.8 value, using a factor = 1 (ie no scaling factor) and with a variable value, fitted to best match the measured and simulated $O_4$ DSCD of each scan. The results of the latter case (variable $O_4$ scaling factor) are used in the following.

MAPA does not perform online Radiative Transfer Model (RTM) simulations but its forward model is provided as pre-calculated look-up tables (LUTs) of differential air mass factors at multiple wavelengths. These LUTs have been calculated offline by a full spherical RTM, McArtim (Deutschmann et al., 2011), following a backward Monte Carlo approach. Since MAPA uses pre-calculated LUTs in combination with a parameterization approach, no a priori values are needed.

The profile inversion relies on finding the minimum differences between measured and modeled DSCDs, leading to the best matching parameters and their confidence interval. MAPA (v0.98) provides various statistics (best match as well as mean and median of the accepted parameter combinations) for the profile parameters as well as for the corresponding profiles. Within the current FRM4DOAS processing chain, only a subset of the MAPA results is included in the output files: the reported VCD are related to the best-match parameters, while the reported profiles are the mean of the ensemble parameters. This has the
drawback that the results for the integrated retrieved profile can be different from the retrieved VCD. However, significant differences of both quantities raise a MAPA warning or error. Additionally, the MAPA retrieved (and reported) VCD considers contributions of the full tropospheric column while the profiles stored in the FRM4DOAS output files stops at 4km. A VCD larger than the reported retrieved profile (from 0 to 4km) can be explained by the contribution of elevated layers above 4km to the VCD, not reported in the common FRM4DOAS output grid, but the opposite case can also happen. This will be changed
in a future version.

MAPA also provides different flags to evaluate the convergence of the algorithm, the consistency of the derived Monte Carlo parameters, and the shape of the profile. More details about MAPA and its flagging algorithm can be found in Beirle et al.





(2019). In the following, we will consider the MAPA data with its own recommended filtering criteria, which is $qa_{mapa}$=0 (ie, keep only valid cases).

The MAPA VCD error are also included in Fig. A1. They are typically a bit smaller than the MMF ones, as they do not include systematic contribution from the HCHO cross-section, see discussion in Sect.A1.

**Table 2.** MAX-DOAS profiling algorithms details.

| Algorithm (version) | MMF (1.0) | MAPA (v098) |
|---|---|---|
| Parameter | | |
| pressure and Temp. | monthly climatology from ECMWF | |
| wavelength | 360nm (O$_4$), 343 nm (HCHO) | |
| simulation grid | up to 60km | na |
| retrieval grid | 200m spacing up to 4km | up to 20km, increasing |
| | | spacing from 5m up to 300m |
| output grid | 200m spacing | 200m spacing |
| | up to 4km | up to 20km |
| surf. albedo | 0.06 | 0.05 |
| aerosol SSA | 0.92 | 0.95 |
| aerosol asymmetry | 0.68 | 0.68 |
| Angstrom exp. | 1 | 1 |
| apriori profile $x_a$ | exp. decay, see eq. (1) | na |
|   - scale height SH | 1km | na |
|   - $VCD_a$ | 8.4 x10$^{15}$ molec/cm$^2$ | na |
| apriori covariance matrix $S_a$ | see eq.(2) | na |
|   - correlation length $\eta$ | 200m | na |
|   - scale parameter SaScal: | 0.5 | na |
| O$_4$ scaling factor | none | (0.8, 1) VAR |

## 2.2  Direct sun DOAS

The Xianghe MAX-DOAS also incorporates a direct sun (DS) viewing mode. The optical head is mounted on a commercial sun tracking system from the Brusag company, equipped with photo-diodes that allow to calculate the sun position, and adjust
to it precisely thanks to a four quadrant. In order to use the same optical elements than in the MAX-DOAS mode, a diffuser plate is inserted before the fiber by means of one of the positions of the filter wheel. The diffuser plate is needed to reduce the direct sun light intensity and to properly match the field of view of the fiber optic. The direct sun mode is included in the normal measurement routine, after the off-axis elevation scans, leading to, in the best cases, a DS measurement every half an hour, if there are no clouds in the field of view.



Unfortunately the filter wheel had some problems after February 2021, and DS measurements over 2018-2022 are only available for the period 11/2019-2/2021.

For the direct sun analysis, the DOAS settings are similar to those used for the MAX-DOAS, but there is no Ring effect, see Table 1. For the retrieval of DS VCDs, slant column densities are first determined with respect to a fixed reference spectrum, and then converted to vertical columns using a geometrical AMF.

$$VCD_{DS} = \frac{DSCD - SCD_{ref}}{AMF_{geom}} \tag{4}$$

One fixed reference spectrum has been used for the whole time-period analysis and the residual HCHO SCD in this spectrum has been estimated using a Langley analysis based on selection of sunny days.

An alternative VCD retrieval was tested by using varying reference spectra, one per season, which improves slightly the DOAS fit residuals, but increases the uncertainty on the VCD columns. We used the difference between these two datasets to
estimate the error on the residual HCHO content in the reference spectrum. The median difference over the whole DOAS direct sun time-period (11/2019 to 2/2021) is of $5.9 \times 10^{14}$ molec/cm$^2$ (about 5.9%), with a linear regression slope and intercept of about 0.85 and $2.5 \times 10^{14}$ molec/cm$^2$ respectively. The random slant column error coming from the QDOAS fit is of the order of 3 to 7% of the VCD up to 60° SZA and around 10 to 12% from 60 to 75°SZA on average. The AMF error is of a few percent (3% maximum), and limited by the non-consideration of refraction effects and a small profile shape dependence at
large SZA. The uncertainty related to the HCHO cross-sections temperature dependence is small, of about 0.05%K$^{-1}$, and can be neglected in our case, as we use the cross-section at 298 K and the bulk of formaldehyde lies in the troposphere. Adding in quadrature these three components (see Fig. A3), we estimate the total error on the direct sun VCD to be about between 10% and 13.5% for the different SZA ranges when considering 6% for the reference spectra and up to 20% if we consider 15% as coming from the difference with respect to an ideal slope of 1.

The DS is typically sensitive to the whole atmosphere, with averaging kernels close to 1 over the whole altitude range, as seen in Fig. 2b. At large SZA, the effect of the Earth sphericity appears, which slightly increases the sensitivity to the surface compared to the highest layers, corresponding to a decrease of the AK down to about 0.8 at 50 km altitude and 80° SZA.

### 2.3   FTIR

Since June 2018, a Bruker IFS 125HR spectrometer is measuring at the Xianghe station, on the roof of the same building
where the MAX-DOAS is operated. The FTIR instrument is described in detail in Zhou et al. (2023). Although the instrument is primary measuring in the near-infrared region (Yang et al., 2020) due to its affiliation to TCCON (Total Carbon Column Observing Network), it also measures in a mid-infrared spectral range, from 1800 to 5500 cm$^{-1}$, allowing the retrieval of many atmospheric components, e.g., O$_3$, CH$_4$, CO, C$_2$H$_2$, C$_2$H$_6$, HCN, NO, C$_3$H$_8$ and HCHO (Vigouroux et al., 2020; Ji and Wang, 2020; Zhou et al., 2021, 2023, 2024; Sha et al., 2021).

The FTIR retrieval principle is also based on OEM. Volume mixing ratio vertical profile information can be derived from the pressure and temperature dependence of the infrared absorption lines. The magnitude of the AK varies depending on the



gas of interest and the chosen fitting micro-windows. If the absorber lines are thin and not too broadened by the atmospheric pressure and temperature, the AKcol can be around 1.

Within the Network for the Detection of Atmospheric Composition Change (NDACC), the HCHO retrieval settings have been harmonized (Vigouroux et al., 2018) to provide a consistent data set among currently 28 FTIR stations, including Xianghe (Vigouroux et al., 2020). The details on the harmonized retrieval settings, also used at Xianghe, can be found in Vigouroux et al. (2018). Among the most important ones are the fitted spectral windows (2763.42–2764.17 cm$^{-1}$; 2765.65–2766.01 cm$^{-1}$; 2778.15–2779.1 cm$^{-1}$; 2780.65–2782.0 cm$^{-1}$) and the spectroscopic parameters used for HCHO and the interfering gases: the so-called atm16 linelist from G. Toon (https://mark4sun.jpl.nasa.gov/toon/linelist/linelist.html). For HCHO, it corresponds to HITRAN 2012 (Rothman et al., 2013). The WACCM v4 model (Garcia et al., 2007) profiles are used as a priori information for the profiles (WACCM averages from 1980 to 2020). A single profile for each FTIR site is used in the time series retrievals. A Tikhonov L1 matrix (Tikhonov, 1963) is used for regularization (Vigouroux et al., 2018).

The HCHO DOFs are limited to 1.0 to 1.6 depending on the station. The mean DOFS for Xianghe in 2020 is 1.32 (and 1.42 for the 1rst of July 2020 example), with a sensitivity mainly located in the whole troposphere as seen in Fig. 2c. The averaging kernels rows are typically spread over several kilometers and the $AK_{col}$ peaks around 10 km and is about 0.8 at the surface (see also Fig. 7), sign that about 80% of the information comes from the retrieval and about 20% from the a priori.

The uncertainty budget has been calculated at each station in Vigouroux et al. (2018) following Rodgers (2000). Depending on the station, the total systematic and random uncertainties of an individual HCHO total column measurement lie between 12% and 27% and between 1 and $11 \times 10^{14}$ molec/cm$^2$, respectively. The median values among all stations are 13% and $2.9 \times 10^{14}$ molec/cm$^2$ for the total systematic and random uncertainties (Vigouroux et al., 2018). For the Xianghe site, the errors on the HCHO VCD are about 3% random and 13% systematic up to 50°SZA, and up to 6% and 16% at larger SZA (up to 75°), as can be seen in Fig. A3.

The Xianghe FTIR data covers the period from June 2018 onward and HCHO data (https://doi.org/10.60897/ndacc.xianghe_ftir.h2co_cas.iap001_rd) are available through the NDACC rapid delivery database (https://www-air.larc.nasa.gov/missions/ndacc/data.html?RapidDelivery=rd-list). The Xianghe FTIR HCHO time-series are used in the TROPOMI validation (Vigouroux et al., 2020) and in on-going Quarterly S5P validation reports (https://s5p-mpc-vdaf.aeronomie.be/), in the GEMS (Lee et al., 2024) and OMI (Müller et al., 2024) validation.

## 2.4 Models

In this paper, we investigate the vertical distribution of HCHO concentrations near Xianghe and evaluate the impact of varying a priori profiles in MAX-DOAS retrievals. To achieve this, we utilize two three-dimensional chemistry transport models: TM5-MP (1° spatial resolution, 30-minute temporal resolution, 34 vertical levels, Williams et al. (2017)) and the CAMS global reanalysis (EAC4, 80km spatial resolution, 3-hour temporal resolution, 60 vertical levels, Inness et al. (2019)). These models are commonly used as a priori in satellite retrievals. Notably, TM5 is used as an input for the QA4ECV products (GOME, SCIAMACHY, GOME-2, OMI) and for the TROPOMI product (which extracts profiles from the TM5-MP forecast). The



CAMS reanalysis profiles have recently been used to generate a consistent ESA Climate Change Initiative (CCI) HCHO climate data record incorporating various satellite sensors. For this study, we consider the monthly averaged profiles from both models interpolated at the Xianghe location and at 9:30 AM and 1:30 PM.

As mentioned above, the FTIR HCHO retrievals rely on a fixed WACCAM model profile as a priori (Vigouroux et al., 2018).

An illustration of the HCHO concentration profiles and VCD columns from the different models are presented in Fig. A4 and Fig. A5. It can be seen that all the models present a decrease of HCHO with altitude, with different rates, depending on the models and the period of the year. The TM5 model has a larger spread in the concentration ranges between the different months of 2020, with smaller values in winter for altitudes between 4 and 20km. This is also reflected in the TM5 total VCD (see Fig. A5) and its contribution above 4km, smaller in winter for TM5 compared to CAMS. The FTIR a priori profile, from the WACCAM average lies in between the CAMS and TM5 monthly profiles and is, by construction, flat in VCD over the year.

## 3   Results and discussion

In this section, we investigate the coherence of the HCHO VCDs and profiles retrieved from the three instruments presented in Sect.2. First we focus on the UV and IR direct sun VCD measurements and then the three MAX-DOAS datasets presented in Sect.2.1 are considered (in Sect. 3.1 and Sect. 3.2 respectively).

### 3.1   VCD comparisons

Figure 1 presents the different VCD datasets considered in this study, focusing on the 2018-2022 period. All datasets present clear seasonality, with an enhanced HCHO signal in summer (up to 25 x$10^{15}$ molec/cm$^2$ on monthly average) and a reduced one in winter (of about 4 to 5 x$10^{15}$ molec/cm$^2$, see also Fig. A9 for the monthly medians VCD). The different instruments have different temporal sampling, with typically a measurement every 20 minutes to half an hour for the MAX-DOAS and the direct sun DOAS (DS) and every hour for the FTIR. The MAX-DOAS instrument also measures under cloudy situations (cloud filtering can be applied in a post-processing step), while DS and FTIR need a clear view of the sun. The different instruments also cover different time-periods, the DS having the smallest data coverage (11/2019-2/2021). We thus focus on that period in the rest of the paper.

### 3.1.1   FTIR vs direct sun DOAS

Figure 3 presents a summary of the FTIR versus UV direct sun (DS) comparisons. It includes quantitative comparisons (statistical regression analysis on scatter plots, histogram and time-series of the absolute differences) and visual comparisons of the diurnal evolution of each dataset, separated per season (including their basic statistics). Both ordinary least square linear (Lin) and Theil-Sen (TS, Sen (1968)) regression statistics are given as inset in the different scatter plots. A summary of the main statistics is also included in Table 3.





**Figure 3.** Comparisons between FTIR and direct sun HCHO VCD data. The first row presents the comparisons of the whole common period (scatter plot, histogram and time-series of the absolute differences), while the second and third rows present the results separated by seasons. The second row is the median diurnal variation (FTIR in black and direct sun in green), with the percentiles 25 to 75 as a shaded area, while the third row presents the seasonal scatter plots with linear and Theil-Sen statistics given as inset. For the median diurnal variation (row 2) a requirement on having at least 10 points per hour has been considered, which explains the lack of curves for March-April-May (MAM), when only 40 comparisons pairs are available (row 3).



**Table 3.** Summary results of the HCHO VCD direct sun (X) vs FTIR and the different MAX-DOAS datasets (Y). The number of comparison points, the total median bias (abs=Y-X and rel=(Y-X)/X) and the median diurnal bias per season are given in the first row, while the Pearson correlation coefficient R and the Theil-Sen regression analysis (slope S, intercept I) for the whole comparisons and for seasonal subsets are given in the following rows. Biases and intercepts I are given in $x10^{15}$ molec/cm$^2$.

| direct sun (X) vs: | | FTIR | MAX-DOAS MMF (flg<2) | MAX-DOAS MAPA (flg<1) |
|---|---|---|---|---|
| nb. | | 878 | 12441 | 6507 |
| median bias: abs, rel | all | -0.49, -6.0% | -2, -21.3% | -2.3, -23.6% |
| diurnal per season | DJF | -0.32, -6.8% | -1.2, -37.5% | -2.5, -70.5% |
| | MAM | n.a., n.a. | -2, -20.6% | -1.3, -15.0% |
| | JJA | -0.92, -3.7% | -3.1, -12.5% | -2.4, -9.3% |
| | SON | -0.87, -9.4% | -3.1, -33.1% | -2.9, -36.2% |
| regression R | all | 0.99 | 0.96 | 0.95 |
| | DJF | 0.97 | 0.92 | 0.72 |
| | MAM | 0.97 | 0.96 | 0.95 |
| | JJA | 0.98 | 0.88 | 0.85 |
| | SON | 0.99 | 0.95 | 0.96 |
| Theil-Sen S | all | 0.97 | 0.88 | 0.97 |
| | DJF | 0.93 | 0.79 | 0.79 |
| | MAM | 0.98 | 0.86 | 0.88 |
| | JJA | 0.99 | 0.93 | 0.98 |
| | SON | 0.97 | 0.92 | 1.02 |
| Theil-Sen I | all | -0.25 | -0.81 | -2.04 |
| | DJF | 0.07 | -0.22 | -1.69 |
| | MAM | 0.37 | -0.39 | -0.03 |
| | JJA | -0.31 | -1.83 | -2.25 |
| | SON | -0.68 | -1.98 | -3.23 |

Data for each data set are selected if available within ±30 minutes of the DS and interpolated on a common temporal grid. It can be seen that the two datasets compare very well, with Pearson correlation R and regression slopes S (both Lin and TS) around 1 for all the periods, small intercepts and very coherent diurnal variations for all the seasons.

The largest discrepancies are found in winter for the slopes (around 0.93 for TS, a bit smaller than the other seasons) and in autumn for the intercept (about -0.68 $x10^{15}$ molec/cm$^2$). The median of all the FTIR-DS differences is -0.5 $x10^{15}$ molec/cm$^2$ (-6%), well within the systematic uncertainty budgets. The largest difference is about -0.9 $x10^{15}$ molec/cm$^2$ (~-9.5%) in autumn (SON).





Unfortunately, the diurnal cycle can only be compared until 14-15 LT due to the lack of FTIR measurements after that time for the following reasons: 1) exhaustion of liquid Nitrogen around 15:30 in the summer time between June - Sep 2020, and 2) a temporary problem in the measurement recording script for HCHO between Oct 2020 and Jan 2021. Moreover, it should be noted that for the diurnal variations subplots, a requirement of at least 10 coincident points per hour has been set, which explains the lack of data for MAM, when only 40 comparison pairs (over 878) are available. This is in part due to the measurement gap in the FTIR dataset in March-May 2020.

Figure A6 shows the diurnal variation panels when the filter is relaxed to at least 5 coincident points per hourly bin. The comparison in MAM is still good, but the diurnal variation is not trustworthy due to the small number of points.

This level of agreement reflects the quality of the instruments, the maturity of the HCHO spectral fitting procedures, as well as the consistency of the HCHO spectroscopic data used in both wavelength ranges. These two datasets are therefore excellent references to test the different MAX-DOAS retrieval strategies. We have chosen the DS as the reference in the following section.

### 3.1.2 MAX-DOAS vs direct sun DOAS

Figure 4 shows the seasonal diurnal variations of HCHO VCD measured by MAX-DOAS compared to those measured by direct sun UV, for the two MAX-DOAS retrieval strategies: (a) MMF and (b) MAPA. For each dataset, data are selected as in Sect. 3.1.1. A summary of the main statistics is also included in Table 3.

It can be seen that all the datasets show similar median diurnal patterns, but with a systematic under-estimation (around -22% on average for the statistics using all seasons, see Table 3) of all the MAX-DOAS VCDs compared to the direct sun VCDs. A striking feature is the different number of coincidences for the two MAX-DOAS datasets, around 3000 each season for the MMF case (where the quality filter is less stringent), to about 1500 to 2000 for MAPA (only the best quality data is retained as default). The median difference is even larger in DJF and SON when HCHO is typically smaller, with differences up to 70% (for MAPA) and 38% (for MMF). In winter there are larger differences in the slopes (multiplicative bias) while in autumn it seems to be more an additive bias, with larger intercepts.

An example of results obtained when selecting only common valid data for both MAX-DOAS algorithms is shown in Figure A7. It can be seen that the common number of valid pairs is drastically reduced, and that the MMF and MAPA results are much closer to each other (by construction of the selection). The effect of the sampling is strongest in winter time for MAPA (from -70.5% to -41.6%) with a change in the diurnal pattern, but the under-estimation of the MAX-DOAS versus DS remains present in all cases.

It should be noted that an under-estimation of the MAX-DOAS columns (being sensitive only in the first kilometers of altitude as discussed in Sect. 2.1) compared to the DS data that are sensitive to the whole atmospheric content (see Sect.2.2, and Fig. 2) is in part expected.

These results are in line with those published in Tirpitz et al. (2021) as part of the second Cabauw Intercomparison of Nitrogen Dioxide measuring Instruments (CINDI-2) where direct sun DOAS HCHO products were higher compared to MAX-DOAS HCHO products. Likewise, recent comparisons of direct sun and MAX-DOAS HCHO measurements from Pandora/PGN in-




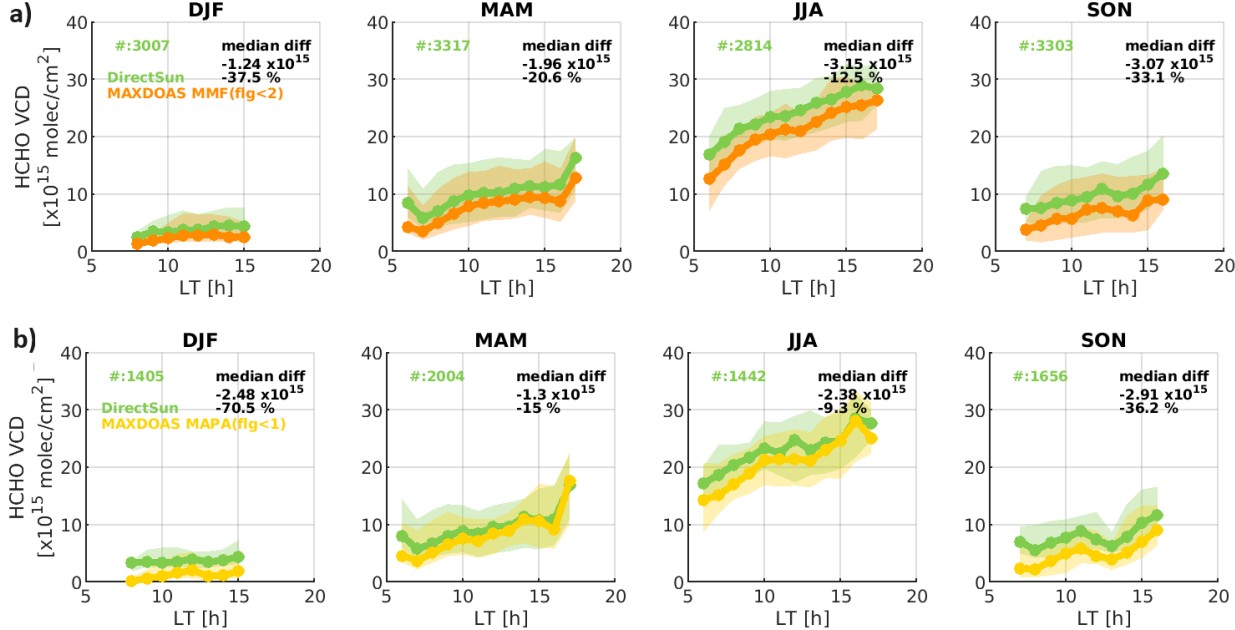

**Figure 4.** MAX-DOAS vs UV direct sun HCHO VCD diurnal variations per seasons, for different MAX-DOAS retrieval strategies: valid (a) MMF results and (b) MAPA results. Only the respective valid data (qa flag <2 and <1 for MMF and MAPA, respectively), as described in Sect. 2.1.1 and 2.1.2, are considered. The number of coincidences for each season are given in each subplot, while the median absolute and relative difference (MAX-DOAS minus DS) are given in the upper right corner of each subplot. A minimum number of 10 points per hourly bin is required.

struments used to validate satellite data over a large number of sites also show larger biases for the PGN direct sun data (for both OMI (Herman and Mao, 2024) and GEMS (Fu et al., 2025)). The difference found in the latter study is larger than what a free tropospheric HCHO could explain, suggesting that, in some sites, the Pandora direct sun product may overestimates HCHO.

Comparisons in Bae et al. (2025) at one site in Incheon (Korea) between collocated MAX-DOAS and Pandora instruments also showed a large over-estimation of the direct sun PGN HCHO data (126%), that was reduced to around 30% after data revision by the PGN team. On the same period, a good agreement of the tropospheric HCHO VCD retrieved from the MAX-DOAS and the Pandora off-axis data was found, in line with previous MAX-DOAS intercomparisons (Pinardi et al., 2013; Kreher et al., 2020; Tirpitz et al., 2021).

**3.2 Profile comparisons**

The next step in the comparison to understand the origin of the VCD under-estimation of the MAX-DOAS shown in Sect. 3.1.2 is to investigate if the observed differences in the columns can be understood by looking at the corresponding differences





in the shape of the retrieved profiles. Indeed, as discussed above, the MAX-DOAS retrievals are very sensitive to the HCHO concentration in the lowest atmospheric layers and optimal estimation retrieval schemes (such as used by MMF) are strongly dependent on the a priori profiles at altitudes above 2 km. The UV DS method is sensitive to the whole atmosphere and does not provide a profile, while the FTIR inversion provides a profile, but with a low associated DOF (1.3 on average in 2020), meaning that the retrieved profile shape is mostly constrained by the a priori profile, coming from the WACCM model.

In a first step, we compare the VCDs and the partial columns up to 4km (pCol4km), and in a second step we compare the HCHO profiles retrieved from the MAX-DOAS and those provided by the FTIR measurements.

Figures A8 and A9 present the monthly averages of the MAX-DOAS and FTIR data for the VCD and the pCol4km. It can be seen that the seasonality is very coherent between the different datasets, with winter/autumn values between 4 and 5 $\times 10^{15}$ molec/cm$^2$ and summer values around 20 to 25 $\times 10^{15}$ molec/cm$^2$. The difference of VCD vs pCol4km is clear in Figure A8, where each dataset is presented separately for more visibility. For the default MAX-DOAS MMF, VCD and pCol4km are the same by construction of the exponentially decreasing a priori, zero above 4km. For the FTIR, the difference is of about 16% over the year, while for the MAX-DOAS differences can vary more over the year, with also negative VCD minus pCol4km for MAPA. In the upper subplot of Fig. A9, the VCDs are compared, while in the lower subplot, the pCol4km are considered. We can clearly see that the retrieved VCDs can differ significantly, with lower columns for MAPA in most months except in summer, while the pCol4km are more coherent along the year, indicating a good consistency of the retrieved MAX-DOAS concentrations in the first 4km, where the technique is mostly sensitive. MAPA pCol4km are smaller in the winter/autumn months (September to December and January to March).

Figure A10 presents in a similar way the monthly median H75 values (i.e. the altitudes where the cumulative partial column reach 75% of the retrieved column profile) and DOFs. It is clear that FTIR data always have a larger H75 value (calculated on the full profile) than the MAX-DOAS ones (calculated on the retrieved profile up to 4km). MAPA results always show smaller values, on average. The smaller the H75 value is, the more the profile is peaked to the surface. For the DOFs, the seasonal sensitivities of the MAX-DOAS and of the FTIR are opposite: the MAX-DOAS is more sensitive in the summer months, when the FTIR has the smallest DOFs.

Figure A11 presents the seasonally averaged MAX-DOAS retrieved profiles from the MAPA and MMF algorithms, for the baseline (only valid flags) and when considering a common selection of points as presented in Sect. 3.1.2 and Fig. A7. The H75 values are also shown in Figure A11. It can be seen that there are substantial differences between the MMF and MAPA profiles, both in terms of profile shape and magnitude, with significant differences for the VCD and pCol4km in winter (DJF) and autumn (SON). In those seasons the H75 are also very different between MMF and MAPA (i.e., 1.7km and 0.5km respectively in winter). The data sampling also strongly affects the VCD, the pCol4km and the H75 values in those cases.

The MMF profiles present some oscillations in summer (JJA), which are likely an indication of the breakdown of the retrieval assumptions and an underestimation of the errors for these conditions. The presence of horizontal gradients or large temporal inhomogeneities within an elevation scan can change the state of the atmosphere and call into question the retrieval assumptions.





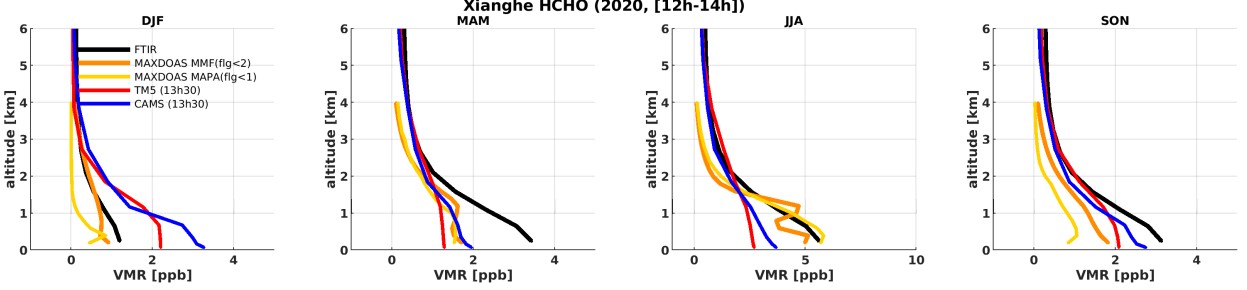

**Figure 5.** Seasonally averaged HCHO retrieved VMR profiles from FTIR (black), MAX-DOAS (MMF and MAPA in different colors) and TM5 (red) and CAMS (blue) models.

If we now focus on each MAX-DOAS algorithm own valid profiles, we can compare MMF and MAPA to FTIR profiles and to CAMS and TM5 models. Figure 5 shows the seasonal averages of all the datasets between 12h and 14h LT. When MMF and MAPA disagree the most (in winter and autumn, as discussed for Fig. A11), MAPA retrieved profiles have a tendency to
decrease faster with altitude. This is also clear in Fig. A11, where the H75 for MAPA is much smaller than e.g. in spring or summer. This partly explains the smaller MAPA columns in comparison to DS data seen in Fig. 4, especially in winter.

The largest difference between MAX-DOAS and FTIR profiles is obtained in autumn (SON), when the FTIR VMRs are larger at all altitudes. This is also true in spring (MAM), but there are only a few FTIR measurements in coincidence with the MAX-DOAS measurements in the March to May 2020 period (cf discussion related to Fig. 3), so this comparison is less
representative.

There is generally a good consistency of the model profiles among them above 1km, with larger differences at the surface. In summer (JJA) and winter (DJF) VMR values from both models are outside of the range of the retrieved VMRs in the first kilometers (respectively lower and larger than the inversions). Both CAMS and TM5 models, and the FTIR profiles, indicate the presence of a significant amount of HCHO above ~4km, where the MAX-DOAS is not sensitive (see Fig. 2 and Fig. 7 and
respective discussions) and where the OE retrieval is strongly constrained by its (exponentially decreasing) a priori. MAPA profiles are also characterized by a rapid decrease with altitude, and they generally yield lowest concentrations in winter and autumn. In this case it is not related to a priori restrictions as MAPA does not rely on a priori estimations, but it can be related to assumptions made in the parametrization scheme and to the fact that, also for MAPA, the sensitivity of the technique is small above 4km of altitude.

The systematically smaller concentrations of the MAX-DOAS retrievals compared to FTIR and to model profiles is clear above ~2 km altitude in all seasons except winter. The underestimation of the MAX-DOAS columns compared to direct sun ones (cf Fig. 4) for those seasons could thus be due to a missing free-tropospheric HCHO column in the MAX-DOAS retrievals. In Fig 6, we evaluate both model data above Xianghe for 2020 and we quantify how much of the total HCHO VCD resides above ~4km. Both models are quite consistent, and suggest that approximately 6 to 10% of the total HCHO columns




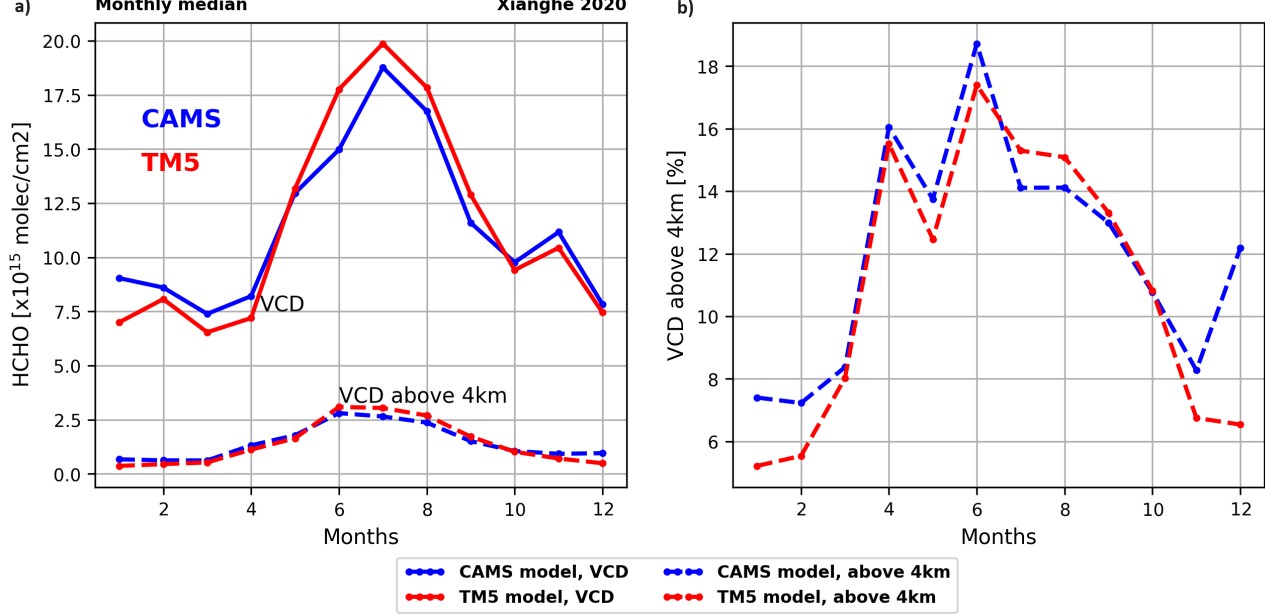

**Figure 6.** Time evolution of the monthly means HCHO VCD from TM5 (red) and CAMS (blue) models over 2020. Both the total VCD (plain lines) and the partial columns of all the layers above 4km (dotted lines) are shown in the panel (a). The panel (b) shows the relative contribution of the upper layers compared to the total VCD.

resides above 4km in autums/winter, and about 12 to 18% in March to September. This explains part of the the gap between MAX-DOAS and direct sun columns.

### 3.2.1 MAXDOAS vs FTIR robust profiles comparison

To quantitatively compare the MAX-DOAS and FTIR retrieved profiles, we need to consider their respective vertical sensitivities. An illustration of the median HCHO column averaging kernels ($AK_{col}$) for the three techniques (MAX-DOAS, DS UV

and FTIR) over the whole 2020 period is given in Fig. 7.

As discussed previously, the sensitivity of the UV direct sun measurements does not depend on the altitude so that AK values are close to 1 over the full range of relevant altitudes. For the MAX-DOAS, the largest sensitivity is maximum close to the surface and decreases relatively fast after 1km of altitude. On average in this case, 100% of the retrieved information comes from the measurement nearby the surface, while at about 2km of altitude 50% of it comes from the a priori. For the FTIR, the

sensitivity is spread throughout the troposphere, with only about 75% of the information coming from the measurement at the surface, and then an increasing dependence on the a priori above the first kilometers, with a tendency to be larger than 1 and peaking around 10km of altitude. Note that this does not affect much the FTIR HCHO VCD retrieval as the typical HCHO



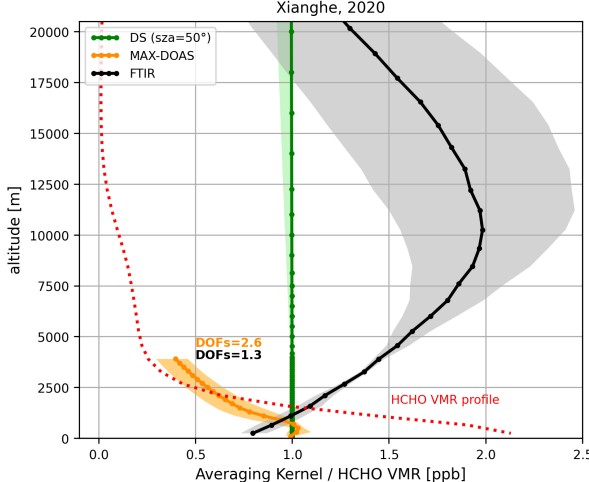

**Figure 7.** MAX-DOAS, FTIR and UV direct sun (DS) HCHO median column averaging kernels and median HCHO VMR profile from FTIR. The median values over 2020 are plotted for FTIR and MAX-DOAS, and the 25-75 percentile range is given as a shaded area.

vertical profile is peaking at the surface, and is rapidly decreasing above 2km, as shown with the red dotted curve in Fig. 7. The typical DOFs (median over 2020) are of about 1.3 for the FTIR and about 2.6 for the MAX-DOAS.

In order to compare MAX-DOAS and FTIR profiles accounting for their respective measurement sensitivities, the method of Rodgers and Connor (2003) is used. It requires the regridding (and extrapolation if needed) of one data set on the altitude of the other, followed by the substitution of the a priori profile.

In our case, the common a priori profile is chosen to be the one used in the FTIR retrievals, therefore the FTIR a priori is substituted in the MAX-DOAS retrieved profile using:

$$x'_M = x_M - (I - A_M) \cdot (x_{M,a} - x_{F,a}) \tag{5}$$

where M stands for MAX-DOAS and F for FTIR. This substituted profile $x'_M$ is then smoothed by the 2nd instrument averaging kernels. They are now both on the same altitude grid.

The MAX-DOAS substituted profile $x'_M$ is smoothed by the FTIR averaging kernels $A_F$:

$$x_{M,smoothed} = x_{F,a} + A_F \cdot (x'_M - x_{F,a}) \tag{6}$$

This approach follows Rodgers and Connor (2003): the highest resolution instrument is smoothed with the lowest one averaging kernels. In our case, the MAX-DOAS profile (substituted with the FTIR a priori) is regridded on the FTIR altitude grid and smoothed with the FTIR AK. In this way, we can reconstruct the MAX-DOAS profile as seen by the FTIR. This is the approach also taken in Vigouroux et al. (2009), Ryan et al. (2020) and Rivera Cárdenas et al. (2021).

As the FTIR profile is defined up to 100km of altitude, we can follow two approaches when regridding and smoothing the

MAX-DOAS profile: a) extending the MAX-DOAS profile up to 100km in order to remain consistent with the FTIR a priori,





or b) only focus on the altitude range from the surface up to 4km, where the MAX-DOAS profiles are originally defined. The statistical results for MAX-DOAS MMF vs FTIR, before and after the substitution and smoothing for the two options, are summarized in Table 4. The median bias and the regression results change significantly.

**Table 4.** Summary of the smoothing tests between FTIR (F, m1) and MAX-DOAS (M, m2). Median relative differences (M-F)/F , Spearman correlation R and Theil-Sen regression slope S and intercept I are given for the original comparisons and after substitution + smoothing. Intercepts are in x$10^{15}$ molec/cm$^2$ and median relative differences in percent.

| | Original | | | | Smoothed | | | | |
|---|---|---|---|---|---|---|---|---|---|
| case name | (M-F)/F | R | S | I | (M-F)/F | R | S | I | description |
| MMF | -20.8% | 0.98 | 0.88 | -0.76 | 0.97% | 0.98 | 0.77 | 2.08 | MMF extended with FTIR prior, then smoothed by FTIR AK |
| MMF with CAMS AP | -17.9% | 0.97 | 0.87 | -0.41 | -1.39% | 0.97 | 0.75 | 2.04 | FTIR vs MMF test CAMS, SaScal=0.4 |
| MMF with TM5 AP | -19.2% | 0.98 | 0.88 | -0.66 | -1.28% | 0.97 | 0.74 | 2.22 | FTIR vs MMF test TM5, SaScal=0.4 |
| MMF, Cut4km | -4.62% | 0.98 | 1.04 | -0.66 | 2.46% | 0.98 | 0.80 | 1.62 | as above but comparison only up to 4km |
| MMF with CAMS AP, Cut4km | -1.75% | 0.97 | 1.03 | -0.35 | -0.1% | 0.97 | 0.77 | 1.6 | as above but comparison only up to 4km |
| MMF with TM5 AP, Cut4km | -3.74% | 0.98 | 1.04 | -0.56 | 0.1% | 0.97 | 0.76 | 1.75 | as above but comparison only up to 4km |

An illustration of the VCD comparison before and after the application of the Rodgers and Conner approach is given in Fig.
8 and in Fig. 9 for the MMF MAX-DOAS dataset.

Figure 8 presents the MAX-DOAS to FTIR original VCD comparisons as a scatter plot. It can be seen that MMF typically underestimates the FTIR VCDs, with a median bias of -20.8%. The correlation coefficient is of 0.98 and the regression parameters are of 0.88 for the slope and -0.76 x$10^{15}$ molec/cm$^2$ for the intercept. The results in Fig. 8b), show that the substitution and smoothing improve the (M-F)/F median bias from around -21% to 1%, conserve the good correlation, but degrade the
regression results, reducing the slope and increasing the intercept, to values around 0.77 and 2.1 x$10^{15}$ molec/cm$^2$.

The comparison of the diurnal variability for the different seasons (Fig. 9) shows that winter (DJF) is the most critical season, when the impact of the smoothing is the largest (about 50% change) leading to larger biases compared to the FTIR columns. This is the season where the MAX-DOAS has the smallest DOFs while it is the largest for the FTIR (around 2.2 for MMF and 1.4 for FTIR vs 2.8 and 1.1 in summer, respectively, see Fig. A10). As discussed earlier (Sect. 2.3), the FTIR data have only
few measurements in MAM, and no coincident data is left for this season when we require at least 10 comparison points to do the hourly average. In summer (JJA) the values are not changed much, except for the first points in the morning, while in autumn (SON) the Rodgers and Conner approach reduces the bias significantly and the diurnal $VCD_{MAX,smooth}$ are in closer agreement to the FTIR than the original MAX-DOAS columns (6.2% vs -29.6%).

When the comparisons are only performed on the 0-4km partial columns, the bias is smaller than for the VCD of about 15%
(this is the typical amount of HCHO above 4 km in the FTIR profiles, see Fig. A8), leading to a (M-F)/F bias of -4.6%, with a correlation of 0.98, a slope of 1 and an intercept of -0.66 x$10^{15}$ molec/cm$^2$. After the smoothing, the overall bias is reduced to 2.5%, but the regression results are again getting worse. Figure A12 shows the MAX-DOAS MMF to FTIR comparisons for



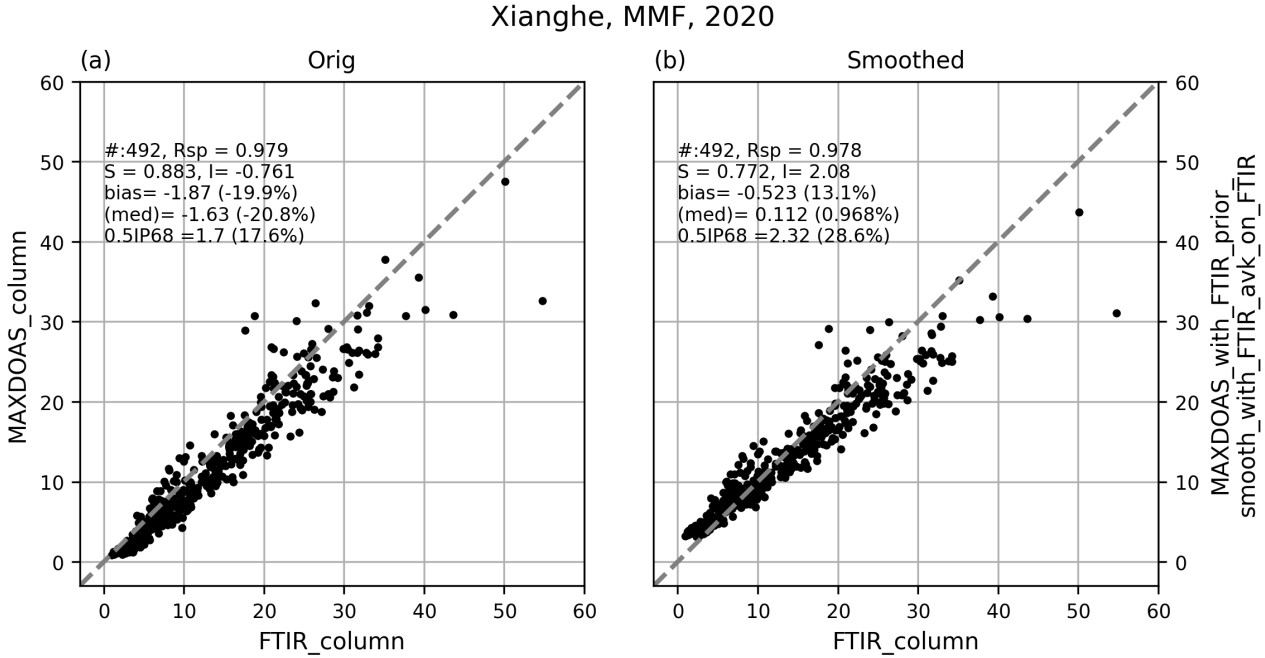

**Figure 8.** FTIR and MAX-DOAS MMF HCHO VCD scatter plot for a) the original data and b) after the substitution and smoothing step as described in the text. Theil-Sen regression statistics, mean and median differences (absolute and relative), as well as its dispersion (as half the interpercentile 68) are given in each panel.

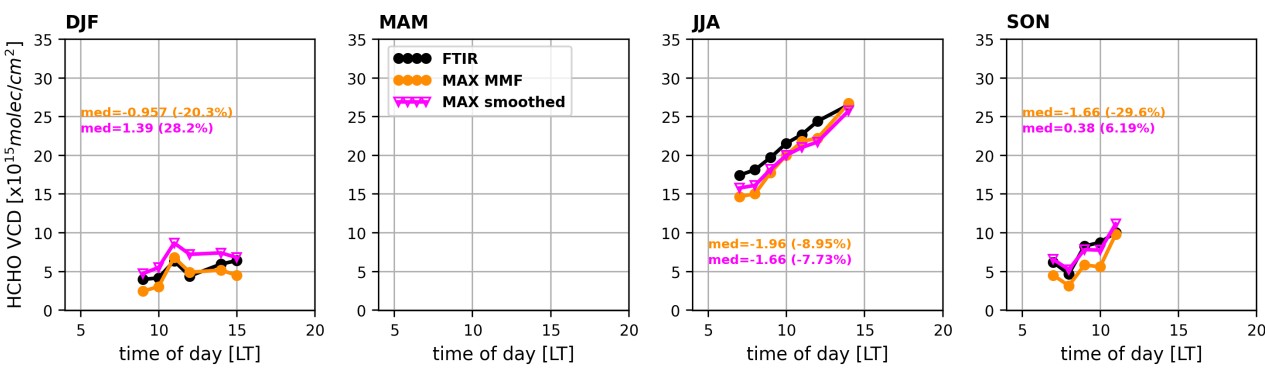

**Figure 9.** FTIR and MAX-DOAS MMF HCHO VCD diurnal variations per seasons. Both the original MAX-DOAS (orange) and the smoothed one (magenta, see text) are shown. Median statistics of the MAX-DOAS minus FTIR absolute and relative differences (M-F)/F are given for each season. A minimum number of 10 points per hourly bin is required.



the different steps of the Rodgers and Connor approach, giving an estimation of which step (regridding, a priori substitution, smoothing) has the largest impact in the final result. It is clear that the reduction of the slope and increase of the intercept

mainly come from the smoothing step, while the regridding to the FTIR grid and the change of the a priori only have a small effect on the regression parameters.

In summary, the MAX-DOAS data are thus under-estimating the total HCHO VCD content, with differences with respect to UV direct-sun data of the order of -22%, and of around -21% compared to the FTIR VCD data. When the comparison is

performed on the 0-4km partial columns, the bias is reduced to -4.6%. When both MAX-DOAS and FTIR averaging kernels and a priori are taken into account, the median bias reduces significantly and is about 1% and 2.5% for the VCD and the 0-4km partial columns.

Compared to MAX-DOAS versus FTIR comparisons available in the literature, we find a similar order of magnitude of the raw (M-F) differences (between 0.3% to 28% depending on the sites). These were conducted in remote mountain-top

observatories (Vigouroux et al., 2009; Franco et al., 2015, for Reunion Island and Jungfraujoch), in remote regions (Ryan et al., 2020, for Lauder and Melbourne) and one in largely polluted area (Rivera Cárdenas et al., 2021) around Mexico City. In those studies, an improvement was also generally obtained when considering the respective vertical sensitivities (Vigouroux et al., 2009; Ryan et al., 2020; Rivera Cárdenas et al., 2021), but the MAX-DOAS instruments had a tendency to report larger HCHO than the FTIR ones, although differences were generally within the combined uncertainties. The mean HCHO levels

are quite different between the different studies: between 0.5 and 2.5 x10$^{15}$ molec/cm$^2$ for the Jungfraujoch, 2-5 to 6 x10$^{15}$ molec/cm$^2$ for Reunion Island, 2.5 ±0.7 x10$^{15}$ molec/cm$^2$ in Lauder, 5.4 ±0.2 x10$^{15}$ molec/cm$^2$ in Melbourne and between 20 to 30 x10$^{15}$ molec/cm$^2$ in Mexico, but with large spatial gradients. Here HCHO columns are typically between $\sim$ 4 to 25 x10$^{15}$ molec/cm$^2$ (see Figure A9) and we expect relatively homogeneous spatial distributions around the Xianghe site due to its suburban nature. Ryan et al. (2020) also focused on the 0–4 km partial columns in his comparison, finding (M-F) monthly

averaged of 15.1 ± 26.3%, and decreasing to 10.1 ± 26.1% when considering the vertical resolutions in the comparison, with correlation R about 0.81 and a linear regression slope of 1.03.

As discussed in the introduction, in recent years the retrieval strategies of both techniques have been harmonized and it is now easier to compare results from different sites, as we rely on coherent inversion choices from different locations. Within the FRM4DOAS consortium, there are four other MAX-DOAS sites (Bremen, Lauder, Toronto and Ny-Alesund, see Van Roozen-

dael et al. (2024)) where FTIR instruments operating in close vicinity, provide HCHO data (Vigouroux et al., 2018, 2020). A preliminary analysis has been performed at those locations , showing similar negative biases of the MAX-DOAS MMF dataset compared to the FTIR NDACC data for the original VCD comparisons, with median differences within -11% to -30%, except in Ny-Alesund, where the HCHO levels are very low. The latter exercise should be extended to account for the different vertical sensitivities and a-priori, to confirm these preliminary findings.

In our comparisons, based on the MMF MAX-DOAS data, we suspect that the underestimation of the original HCHO VCDs compared to direct sun results, is mostly due to the lack of sensitivity of the MAX-DOAS above 4km, along with the too fast decrease with altitude of the chosen exponential decreasing a priori profile for the MMF MAX-DOAS retrieval. This is





reduced when applying the Rodgers and Conner approach. The current choice of the MAX-DOAS a priori means that the free-tropospheric HCHO content is not accounted for. To further test this hypothesis, we run the MMF algorithm with different
a priori profiles.

### 3.2.2 Change a priori profile in MMF processing

CAMS and TM5 models profiles shown in Fig. 5 have been used as a priori for the MMF MAX-DOAS retrievals, over the whole year 2020. The different tests start from the same DSCD and aerosols retrievals, and only recalculate retrieved HCHO concentrations with some changes in the inversion settings. In addition to the change of the HCHO a priori (from the
exponential decrease of Eq. 1 to the monthly means CAMS or TM5 concentrations), the a priori constraint (the $S_a$ covariance matrix) can also be adapted. As in this MMF implementation the $S_a$ matrix is constructed from the a priori profile itself (see Eq. 2) we have tested several scaling parameters in order to keep similar $DOFs$ while disentangling the effect of the a priori shape change from the induced effect of (over)constraining the prior.

Different Sa scaling parameters have been tested for a few months and the best candidate to keep similar $DOFs$ is a $S_a$ scaling factor (SaScal) of 0.4 (instead of the 0.5 in the original FRM4DOAS MMF implementation, see table 2).

Figure 10 presents the median retrieved profiles and the cumulative partial column DOFs for the months of February and July 2020 for the original FRM4DOAS MMF data (in orange) and two other tests with the TM5 (in red) and CAMS (in blue) profiles as a priori. Both the retrieved quantities (solid lines) and the a priori concentration profiles (dotted lines) are shown. In
winter, we can see that the retrieved profiles are very similar for the three options below 1km (where we have most information), while they diverge above, where they follow the different a priori constrains. The contribution to DOFs from layers above 2km tends to zero. In summer, the use of model a priori profiles also lead to very coherent inversions in the first layers for the three cases, as well as a removal of the oscillation in the retrieved profile (between 700m and 1.5km), that was present in the original MMF inversion. The larger concentrations of the model a priori in elevated layers compared to the exponentially
decreasing profile originally used in MMF result in larger retrieved values above 1.5km. In the original inversions, this larger HCHO concentration was somehow pushed between 500m and 1.5km, where the gain in freedom (about 0.7 increase in the cumulative partial column DOF) allowed the retrieval to deviate from the a priori. As a result, an oscillation was produced.

Above 2km, the DOFs becomes larger than originally retrieved. For both winter and summer examples, we can see the rapid increase of the cumulative partial column DOFs in the first layers, where the retrievals are diverging from their a priori profiles,
while above 1.5-2km, each retrieved profile is constrained to its prior, and the cumulative DOFs tends to a constant value. By construction, the HCHO concentration above 1.5km for both CAMS and TM5 tested cases are therefore larger compared to the original MMF retrieval, leading to slightly larger simulated VCDs (around $2.4 \times 10^{15}$ molec/cm$^2$ instead of $2 \times 10^{15}$ molec/cm$^2$ in summer) and only small changes in the total DOFs ($\sim$2.9 compared to 2.77 in summer). It should be noted, however, that the 0-4km partial columns (summing the retrieved profiles from 0 to 4km), are smaller, as the retrieval is only performed in
the 0-4km altitude range, while the given a priori is defined on the whole atmosphere, up to 60km, defacto adding an effective ghost column to the reported VCDs.





**Figure 10.** MAX-DOAS a priori (dotted lines) and retrieved (plain lines) profiles (left panel) and corresponding cumulative partial column DOFs (right panel) for the month of February and July 2020 for MMF (in orange), TM5 apriori test case (in red) and CAMS apriori test case (in blue). The median values are shown with the addition of a shaded area representing the 25 to 75 percentiles values. The altitude is given in meters above the surface of the center of the retrieval layers.

Both TM5 and CAMS model profiles, when used as a priori with a SaScal=0.4, lead to similar results in terms of retrieved DOFs. A full year has been analyzed for those cases (see Fig. A8-A10). To quantify the impact of the change in a priori profile, we can compare the VCDs from the different sensitivity tests, to the direct sun VCD, as previously done in Fig. 4. Results are

shown in Fig. 11. A large improvement in the MAX-DOAS minus DS absolute differences is obtained when using the CAMS or TM5 profiles as a priori instead of the exponentially decreasing shape of the default MMF case. The reduction of the bias of



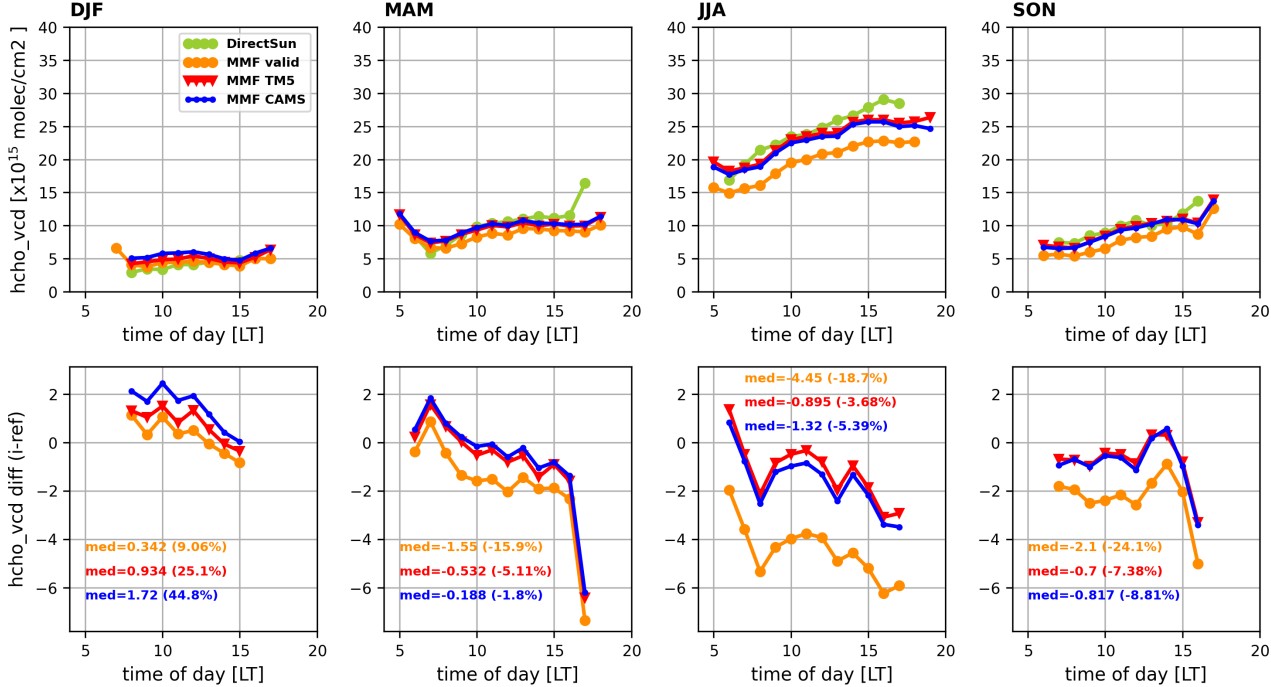

**Figure 11.** Direct sun and MAX-DOAS (MMF in orange, TM5 apriori test case in red and CAMS apriori test case in blue) HCHO VCD diurnal variations per seasons (upper panels) and their differences (M-D, lower panels). Median statistics of the MAX-DOAS minus direct sun absolute and relative differences (M-D)/D are given for each season in the corresponding color. A minimum number of 10 points per hourly bin is required.

the MAX-DOAS VCD with respect to the direct sun VCD is clear in all seasons except winter, when the difference between using the CAMS or the TM5 models as a priori is the largest, leading to VCDs larger than the direct sun ones. The differences are not constant during the day, and increase for larger SZA, when the pointing directions of the MAX-DOAS and the direct

sun modes are increasingly different. A similar comparison is done in Fig. A13 for MAX-DOAS pCol4km versus DS VCD, and only significant impact of the CTM a priori tests is found in winter, leading to larger differences. In winter, the larger discrepancies therefore seem to come from the HCHO contribution below 4 km of altitude, while in the other seasons, it is the free tropospheric content included in the VCD that leads to the largest impact.

The change in the MAX-DOAS profile shape, can be estimated through the H75 parameter. The monthly averaged compar-

isons are shown in Fig. A10. Both CAMS and TM5 have the tendency to increase the H75 values for most of the seasons, i.e., profiles less peaked to the surface.

Fig. 12 presents the diurnal variation per season, but this time with respect to FTIR data, also including the effect of using the Rodgers and Conner approach on the vertical profiles distributions (as described in Sect. 3.2.1). The FTIR (black), the MAX-DOAS (blue) and the smoothed (cyan) MAX-DOAS MMF case with the CAMS monthly profiles as a priori and a





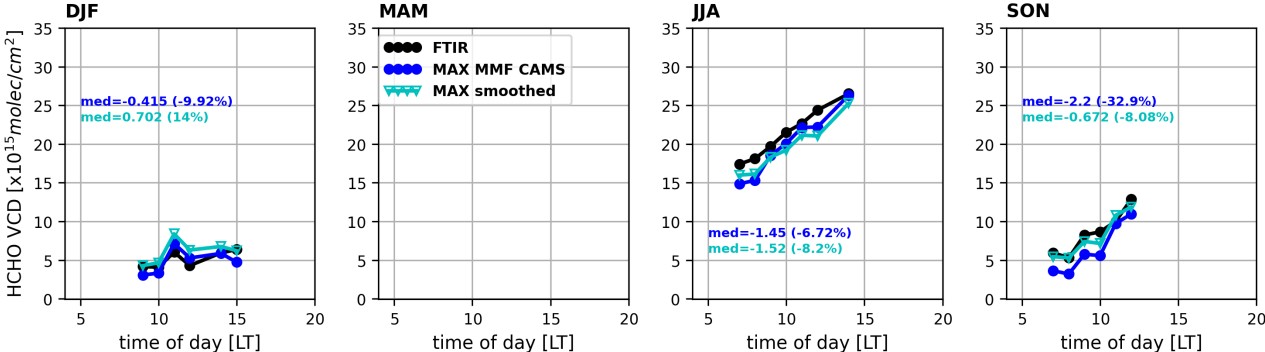

**Figure 12.** FTIR and MAX-DOAS (CAMS apriori test case) HCHO VCD diurnal variations per season. Both the original MAX-DOAS CAMS (blue) and the smoothed one (cyan) are shown. Median statistics of the MAX-DOAS minus FTIR absolute and relative differences (M-F)/F are given for each season. A minimum number of 10 points per hourly bin is required.

scaling parameter SaScal=0.4 are shown. This figure is similar to Fig. 9 for the original MMF inversion. Table 4 also presents the comparisons of the two sensitivity tests, before and after the regridding/smoothing. From Fig. 12 and Table 4 it can be seen that over the whole year, the test with the TM5 and CAMS models as a priori leads to slightly smaller total bias (-19.2% and -17.9% instead of -20.8% with MMF), which all reduce to less than -1.4% after the a priori substitution, regridding and smoothing, but again with worse regression statistics (smaller slopes and intercept around $2 \times 10^{15}$ molec/cm$^2$). A positive bias

is also clear in winter time from Fig. 12 (and Fig. A14 for the TM5 case), as it was already present in Fig. 9 for the original inversion.

It should be noted that the diurnal variation plots are made with some filtering: only hourly bins with at least 10 comparison pairs for the season are kept, in order to have meaningful binning of the hourly medians. The median biases given as inset in these diurnal plots per season can thus be different than what found in Table 4 and in the scatter plots, where all the comparison

pairs are kept.

In order to investigate the remaining winter over-estimation of the MAX-DOAS, an additional test for CAMS a priori but with a bit more of freedom (SaSCal=0.6) is shown in Figure A15 for December 2020. It can be seen that the median retrieved profile is slightly smaller than the one with SaSCal=0.4, with a larger range of variability and a smaller monthly VCD (3.55 $\times 10^{15}$ molec/cm$^2$ instead of 3.81 $\times 10^{15}$ molec/cm$^2$), similar surface value than FRM4DOAS, smaller concentration up to 2km

and then larger concentration than FRM4DOAS but smaller than the previous test above. The DOF (=2.14) is also closer to the original MMF one (2.19) than the previous test (1.92).

It should be noted that for the comparisons of MAX-DOAS profiles with FTIR, the impact of the change in the MAX-DOAS a priori is smaller than when looking at VCD, as for Fig. 11 for DS. This is related to a discrepancy for the new tested



cases, between the retrieved profiles (inversed only on the 0-4km retrieval grid range) and the reported VCD (estimated on the whole atmosphere, including also contribution of the a priori above 4km). The 0-4km partial columns (calculated integrating the profile) are then smaller than the reported VCD, that include a significant contribution from the a priori above 4km. An illustration of the difference between VCD and pCol4km for each case is shown in Fig. A8 over 2020. It is relatively constant around 16% for FTIR, it is zero by construction for MMF, and it varies around 13% and 12% on average for the MMF retrievals

with CAMS or TM5 as a priori.

If we focus the MAX-DOAS to FTIR profiles comparisons only on the 0-4km range, as also done in Sect. 3.2.1 and reported in Table 4, it can be seen that with this configuration, the change of a priori has a different impact on the original bias (-1.75% using CAMS and -3.74% using TM5, compared to -4.6% for the original MMF case) while using the Rodgers and Conner approach is reducing the median bias over 2020, to values around -0.1% and 0.1% respectively.

The regression parameters are similar for the three cases and in line with results obtained before when considering the whole FTIR grid for the comparison. Slopes around 1 with the smallest intercept for the original comparison are obtained with the CAMS a priori profile, and a decrease of the slope and an increase of the intercept to values close to 1.6 x$10^{15}$ molec/cm$^2$ after the substitution and smoothing.

## 4    Summary and conclusions

In this paper, we investigated HCHO VCD and profile measurements performed over Xianghe between mid 2018 to the end of 2021, using three different instrument types: MAX-DOAS, UV direct sun and FTIR. These instruments have different sensitivities to the HCHO altitude distribution, and these have been taken into account in the analysis. For the MAX-DOAS, MMF and MAPA datasets have been considered, as implemented in the FRM4DOAS centralized processing system.

Comparing the direct sun datasets, an excellent agreement of the FTIR and direct sun DOAS HCHO VCDs is found, with a

correlation of 0.99, a slope of 0.97 and a median bias of -5 x$10^{14}$ molec/cm$^2$ (-6%), well within the combined uncertainties. The diurnal cycles for the different seasons are also in close agreement, and these data sources therefore provide a ideal reference for comparison with the MAX-DOAS HCHO datasets.

Generally, the MAX-DOAS column and profile retrievals from MMF and MAPA algorithms are in good agreement, with a good correlation (0.96) against DS data and coherent seasonal and diurnal patterns, but they both systematically underestimate

the direct sun DOAS and FTIR VCDs. The median bias over the period is of about -22%, with the MAX-DOAS always being smaller than the DS. This underestimation is larger in DJF and SON.

When comparing the MAX-DOAS profiles to the FTIR data, a negative bias of about -20.8% is also found. However, when applying the rigorous Rodgers and Connor approach, harmonizing the a priori profiles and smoothing the MAX-DOAS data with the FTIR kernels, the agreement improves to 1% on average, but also leads to worse regression parameters, with a reduced

slope and an increased intercept. The smoothed comparisons are deteriorated in winter. When the comparison is performed only on the 0 to 4km altitude grid, the original median bias is reduced to -4.6%, and to 2.5% after smoothing.





The underestimation of the MAX-DOAS is attributed to the lack of sensitivity of the MAX-DOAS technique above 2km altitude, along with the unrealistic a priori profile used above that height in the retrieval. As a result, the free-tropospheric HCHO content reported by the models and observed by the direct sun UV and IR measurements is not detected by the MAX-DOAS measurements. When changing the MMF a-priori exponentially decreasing profile with modeled monthly mean profiles, the new MAX-DOAS datasets show a much better agreement with direct sun DOAS and FTIR data. Using CAMS or TM5 as a priori, and testing different prior constrains change slightly the results, with some dependences on the season. The model a priori profiles have a positive impact on VCD in all seasons except in winter, when the VCD comparisons are degraded. When considering the pCol4km, the impact of the CTM a priori is only significant in the winter period. The VCD improvement is thus mainly coming from the included free-tropospheric content, smaller in winter. The prior substitution and smoothing effect for the MAX-DOAS to FTIR comparisons, is very small in summer.

Introducing model a priori profiles as experimented here corresponds to adding a ghost column to the retrieved MAX-DOAS HCHO columns, which has the advantage to close the gap between the MAX-DOAS VCD retrievals and the direct sun measurements. Another (or complementary) option could be to provide the HCHO partial column corresponding to the layers having the largest sensitivity (as assessed through the averaging kernels).

The MAX-DOAS HCHO MMF and MAPA datasets run within FRM4DOAS show consistent diurnal and seasonal behaviors compared to the other datasets. Some aspects should however be improved and we studied the impact of changing the MMF a priori profile to a more realistic one. The current under-estimations in winter of MAPA and the slopes around 0.8 with positive intercept of MMF versus FTIR (especially after applying the Rodgers and Connor approach) should however be better understood.

The results shown in this paper using TM5 and CAMS assume that the models are good in the free troposphere. This assumption seems to be coherent above Xianghe, but more work might be needed to validate the modeled HCHO concentrations in the global free troposphere. Getting information on the HCHO profile with airborne in-situ measurements above MAX-DOAS sites campaign (such as with the In Situ Airborne Formaldehyde instrument (ISAF, Cazorla et al., 2015) or the COmpact Formaldehyde FluorescencE Experiment (COFFEE, St. Clair et al., 2017), as recommended in Merlaud et al. (2020) would help in quantifying the free tropospheric HCHO content. Comparison of these MAX-DOAS results to MAX-DOAS network measurements performed in China (Song et al., 2023; Jiao et al., 2025) and to HCHO surface concentration measurements (e.g., with proton transfer reaction mass spectrometer, PTR-MS, Wei et al., 2023) would be beneficial to further validate the MAX-DOAS HCHO data.

Reaching a consolidated FRM4DAS HCHO approach for all the sites is a necessary step forward for the network of MAX-DOAS instruments capable of detecting HCHO. Indeed, the lack of harmonization of the HCHO VCD inversions and profiles retrievals is currently considered as a limiting factor in its use as a consolidated network, e.g. for satellite validation (see e.g. De Smedt et al. (2021).

In the current implementation of the FRM4DOAS project, 18 sites are already included in the HCHO processing with harmonized DOAS and parallel MAPA and MMF exponentially decreasing a priori profile retrieval settings (Van Roozendael et al., 2024). About twenty sites confirmed their interest to be included in future extension and automated operations.





Moreover, around five MAX-DOAS instruments of the current FRM4DOAS consortium are located in sites where an FTIR is also measuring and a work similar to what presented in this paper for Xianghe would be a nice extension to confirm our findings. Intercomparison of ground-based MAX-DOAS, direct sun (like PGN) and FTIR instruments is a key requisite to exploit their data in an integrated and harmonized way.






## Appendix A: Datasets characterization

### A1    Errors estimation

Figures A1 and A2 presents the different contribution to the HCHO VCD errors, as the percentage value of the VCD columns, as provided in the netCDF MAX-DOAS FRM4DOAS 01.01 files and in the FTIR GEOMS files available from NDACC RD.

The median value within 5° steps SZA bins from 15° to 80° are shown as box whisker plots.

For the FTIR data, the errors are provided as separated in random and systematic contributions, with typically a larger contribution from the systematic part of the errors. In the MAX-DOAS current internal netCDF FRM4DOAS files, the total VCD errors are provided for both MMF and MAPA, without separation of random and systematic contributions. An illustration of the error estimation for MMF and MAPA valid data is given in Fig. A1. It can be seen that the MMF errors are always larger,

as they also include a 9% systematic contribution to the errors as coming from the uncertainty on the HCHO cross-section (see e.g., Pinardi et al., 2013), which is not included in the MAPA errors. The MAPA VCD errors are estimated from weighted standard deviation of all matching profile parameters c that equals the VCD. The MMF errors are estimated from the covariance smoothing error matrix and the covariance measurement noise error matrix and include a systematic contribution as a fixed fraction of VCD (9% for HCHO).

Figure A3 summarize the errors for each technique. Systematic constant errors are not displayed in this figure, but they are considered in quadrature when calculating the total contribution for each dataset. For DS this includes 3% AMF uncertainty, 6 to 15% reference spectra contribution (leading to the two different total curves for the DS).However, for the MAX-DOAS, these values are likely underestimated. E.g., for MMF, the current DOAS error contribution only includes an estimation from the DSCD fit and a systematic contribution for the HCHO cross-section uncertainty (9%, from Meller and Moortgat (2000)),

while we know that other DOAS fit parameters play a significant role in the DSCD error, as discussed in Pinardi et al. (2013) Figure 18.

### A2    Vertical profiles and seasonality

Figure A4 presents the model concentration profiles (CAMS, TM5 but also the WACCAM profile used as a priori in the FTIR retrievals). Profiles up to 40km are shown in the upper row and the lower row is a zoom in the troposphere, up to 10km. It can

be seen that both CAMS and TM5 present similar seasonal variability and similar altitude dependency. The TM5 is however decreasing a bit faster vs altitude for the winter months.

Figure A5 summarizes the models and the FTIR total VCD and the contribution of the partial columns above 4km. The models present a seasonality in the relative contribution of the part above 4km (from 6% to about 18% in summer), while the retrieved FTIR is more constant over the seasons, with a contribution around 15%.

Figures A8 and A9 presents the monthly averages of the MAX-DOAS and FTIR data for the VCD and the partial columns between and 4km (pCol4km). It can be seen that the seasonality is very coherent between the different datasets, with winter/autumn values between 4 and 5 $x10^{15}$ molec/cm$^2$ and summer values around 20 to 25 $x10^{15}$ molec/cm$^2$. The difference of VCD vs pCol4km is clear in Figure A8, where each dataset is presented separately for more visibility. For the default





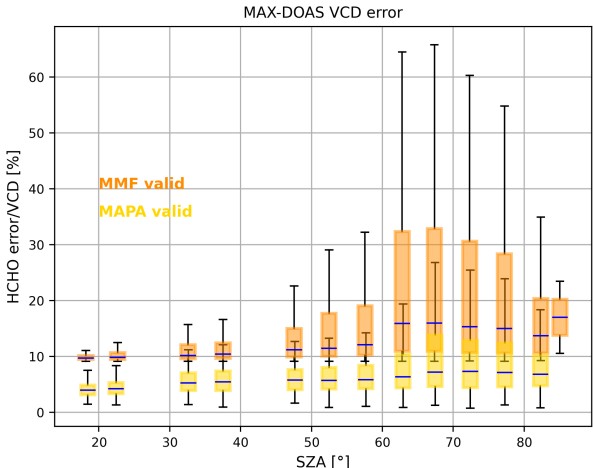

**Figure A1.** MAX-DOAS HCHO VCD errors as a function of SZA bins (for both MMF and MAPA valid points).

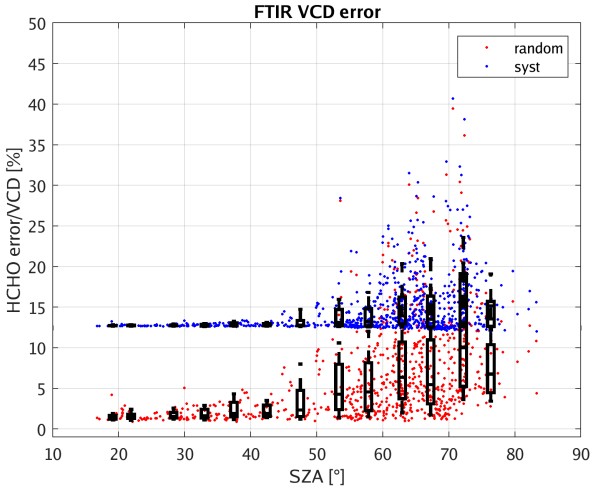

**Figure A2.** FTIR HCHO VCD errors as a function of SZA bins (only DS measurement period).

MAX-DOAS MMF, VCD and pCol4km are the same by construction of the exponentially decreasing a priori, zero above 4km. For the FTIR, the difference is of about 16% over the year, as it is also clearly visible in FigA5. For the MAX-DOAS the contribution varies depending on the algorithm, but are generally smaller and more variables over the year.

In the upper subplot of Fig. A9, the VCD are compared, while in the lower subplot, the pCol4km partial columns are considered. We can clearly see that the retrieved VCD can differ significantly, with lower columns for MAPA in most months except in summer, while the pCol4km are more coherent along the year, sign of good consistency of the retrieved MAX-DOAS concentrations in the first 4km, where it is sensitive. MAPA pCol4km are smaller in the winter/autumn months (September to





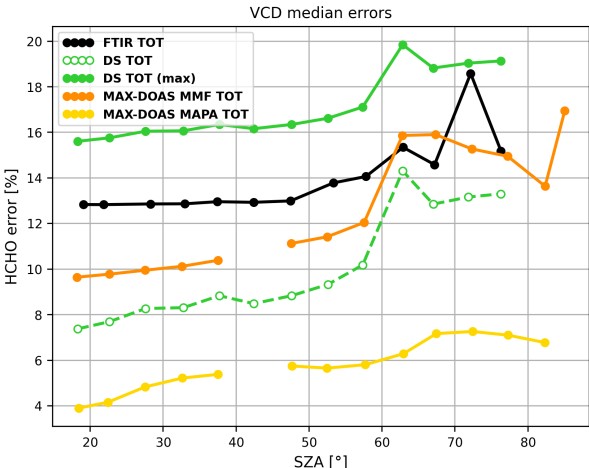

**Figure A3.** Direct sun (DS), MAX-DOAS (MMF and MAPA) and FTIR HCHO median VCD errors as a function of SZA bins. Systematic constant errors are not displayed in the figure, but they are considered in quadrature when calculating the total contribution for the DS dataset. This includes 3% AMF uncertainty, 6 to 15% reference spectra contribution (leading to the two different total curves for the DS). For the DS dataset, these are numbers estimated within this work, while for FTIR these are the numbers reported in the GEOMS harmonized files format and for the MAX-DOAS in the FRM4DOAS netCDF files.

December and January to March). More robust comparison, taking into account coincident points and vertical sensitivities is presented in Sect. 3.2.1.

Figure A11 presents the seasonally averaged MAX-DOAS retrieved profiles and AKcol (when relevant) from the MAPA and MMF algorithms, for the baseline (only valid flags) and when considering a common seelction of the points based on a common vaildty flag, similar to what done for $NO_2$ in the FRM4DOAS official product (see discussion in Sect. 2.1 and Van Roozendael et al. (2024)).

The FRM tests with the different monthly mean model profiles as retrieval a priori have relatively similar results when comparing their results to the FTIR HCHO data. Figure A14 is the corresponding figure to Fig. 11. We can see that the difference between the two model cases is essentially in winter time, where the retrieved MAX-DOAS columns starting from the CAMS a priori profile are about 10% closer to the FTIR ones (about -10% smaller) before smoothing, and a bit larger than FTIR after smoothing (but about 10% smaller than when starting from the TM5 a priori). As for Fig. 11, the hourly bins have been considered only if there were at least 10 comparison pairs/points for that hour.







**Figure A4.** HCHO concentration profiles over Xianghe for FTIR a priori (WACCAM model) and for each month of CAMS and TM5 models in 2020. The second row is a zoom up to 10km of the first one.



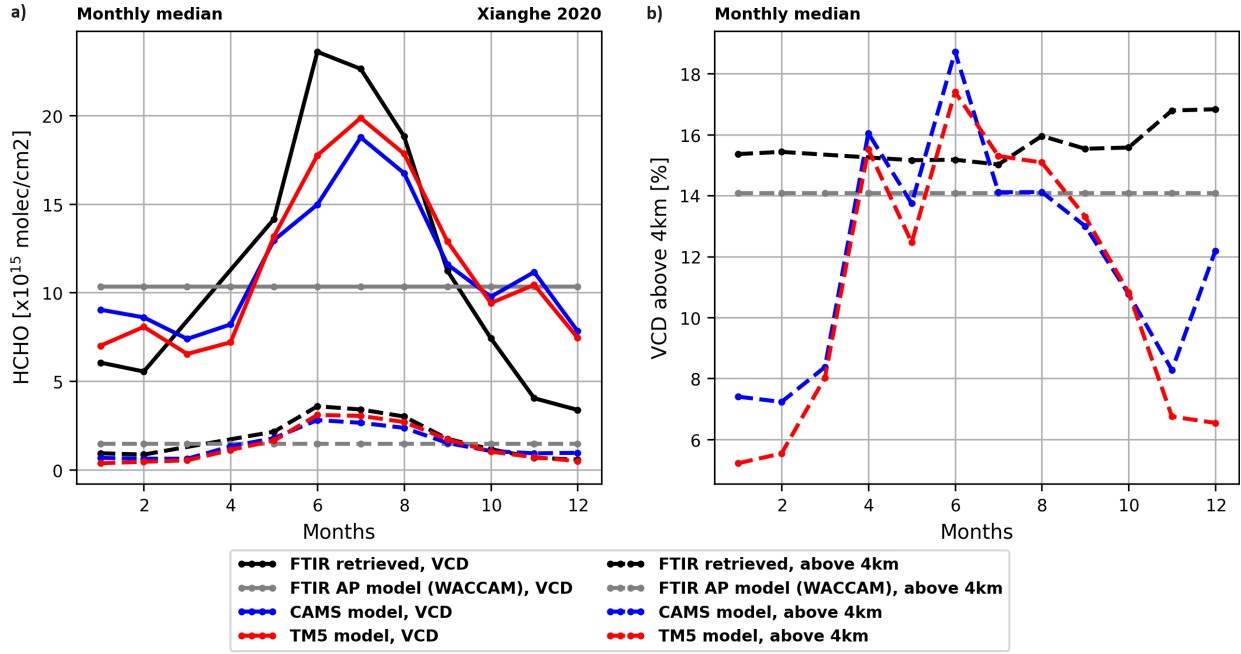

**Figure A5.** Time evolution of the monthly means HCHO VCD from FTIR a priori (gray), FTIR retrieval (black) and TM5 (red) and CAMS (blue) models over 2020. Both the total VCD (plain lines) and the partial columns of all the layers above 4km (dotted lines) are shown in the panel (a). The panel (b) shows the relative contribution of the upper layers compared to the total VCD.

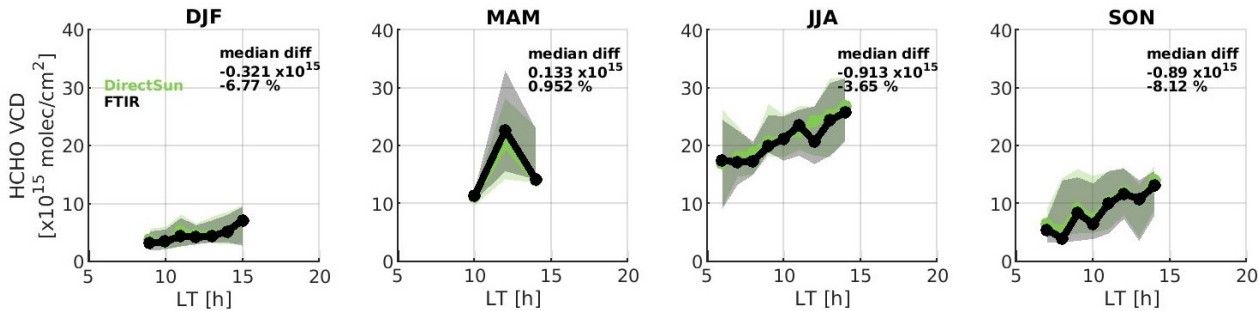

**Figure A6.** As second row of Fig. 3, but only requiring at least 5 points for hourly bins.



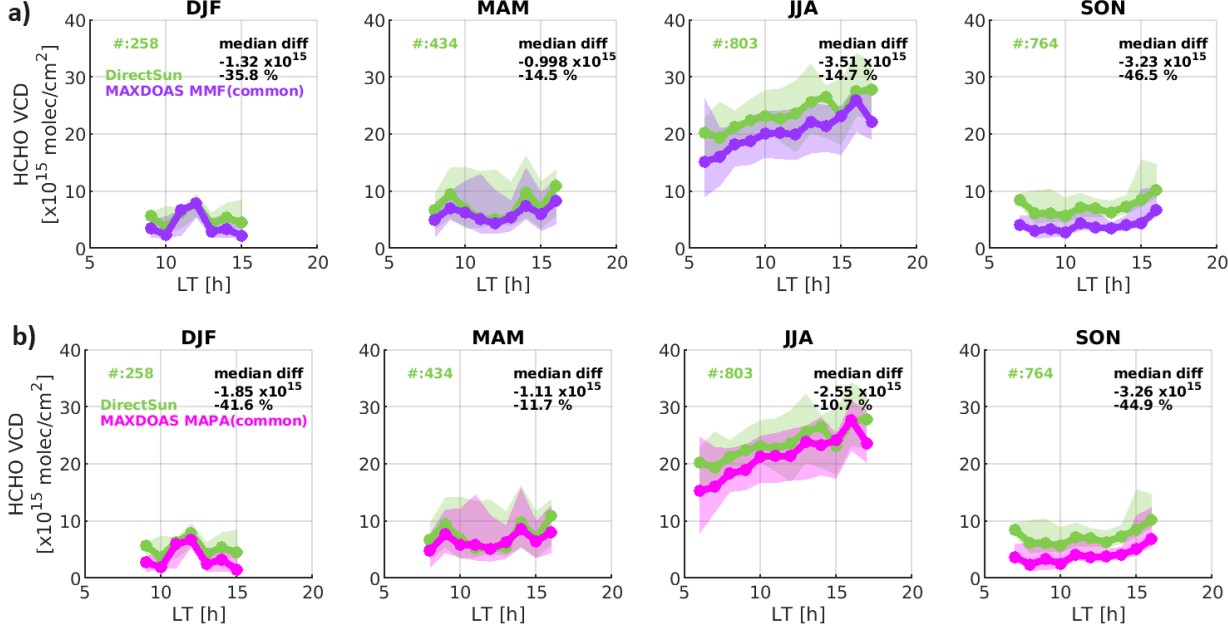

**Figure A7.** As Fig. 4, but only for the common MMF and MAPA valid points.

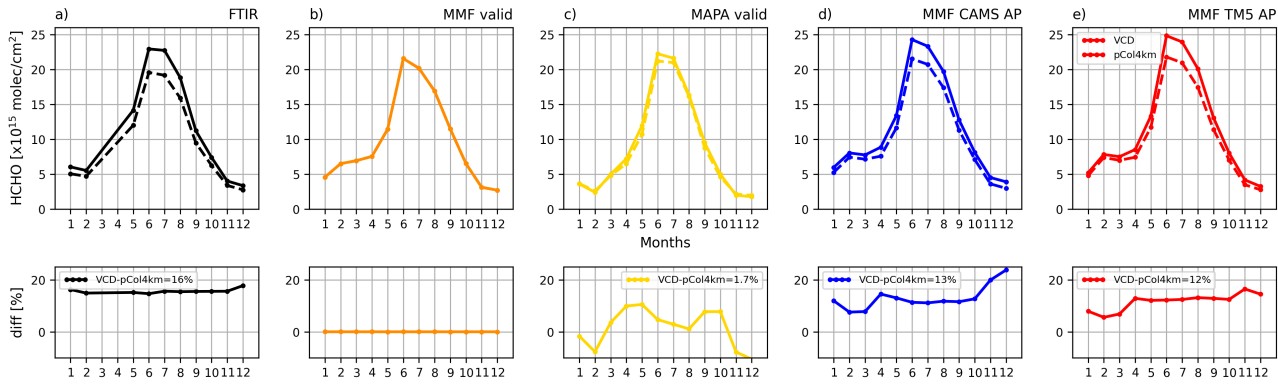

**Figure A8.** Monthly averaged HCHO VCD and 0-4km partial columns from FTIR and MAX-DOAS MMF, MAPA and when using CAMS or TM5 as a priori profiles and their differences.



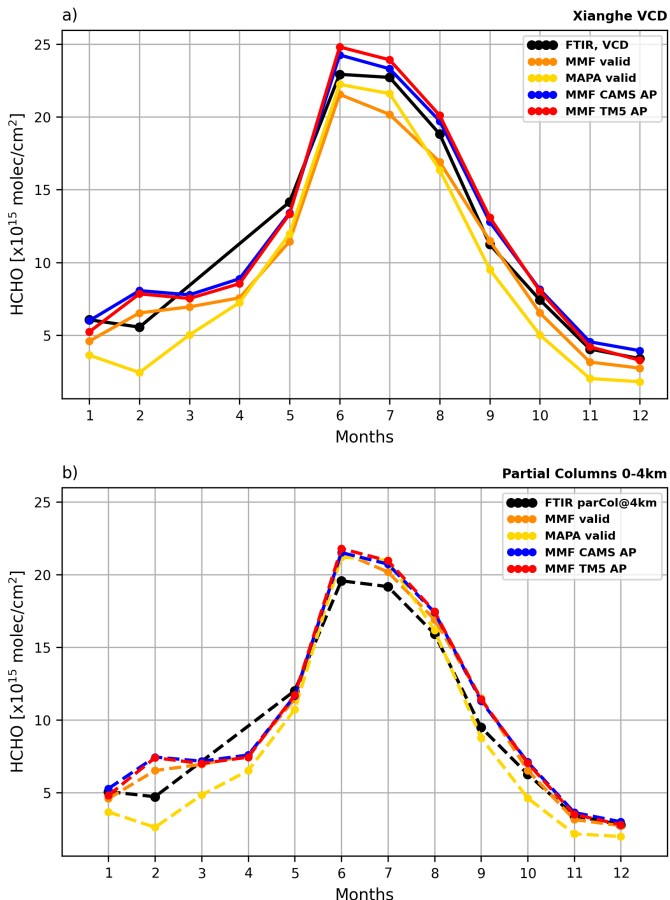

**Figure A9.** Monthly averaged HCHO a) VCD and b) 0-4km partial columns from FTIR and MAX-DOAS MMF, MAPA and when using CAMS or TM5 as a priori profiles.



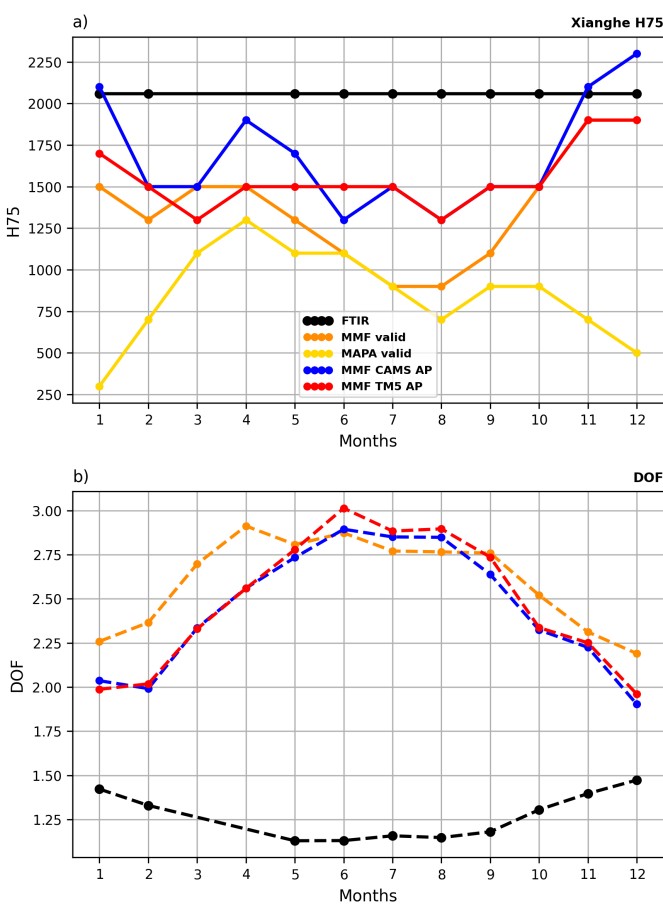

**Figure A10.** Monthly averaged HCHO a) H75 and b) DOF (when relevantà from FTIR and MAX-DOAS MMF, MAPA and when using CAMS or TM5 as a priori profiles.



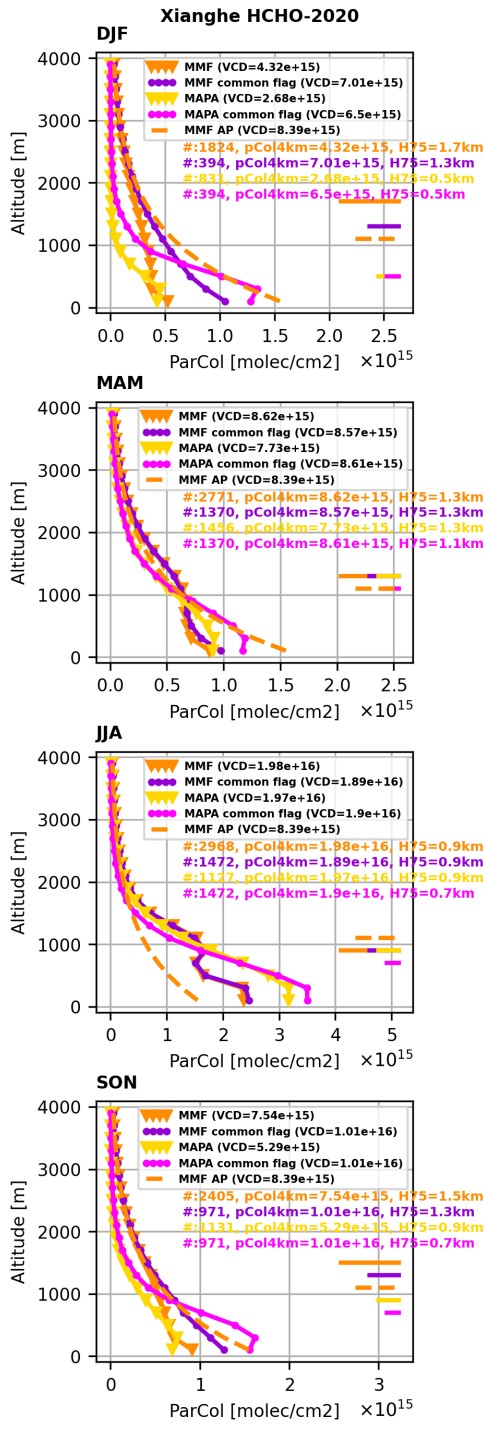

**Figure A11.** Seasonally averaged HCHO profiles from MAX-DOAS MMF and MAPA valid points and when keeping only common selection.





**Figure A12.** FTIR and MAX-DOAS HCHO VCD scatter plot for the different steps of the Rodgers and Connor approach: a) the original comparisons, b) when the MAX-DOAS is regridded on FTIR grid, c) when the MAX-DOAS a priori is replaced by the FTIR one, and d) when all the steps are included (prior substitution and smoothing with FTIR averaging kernels). The plots are based on the MMF case when cutting the comparison at 4km. The median bias and the regression statistics are given in each subplot. The months of 2020 are color coded.





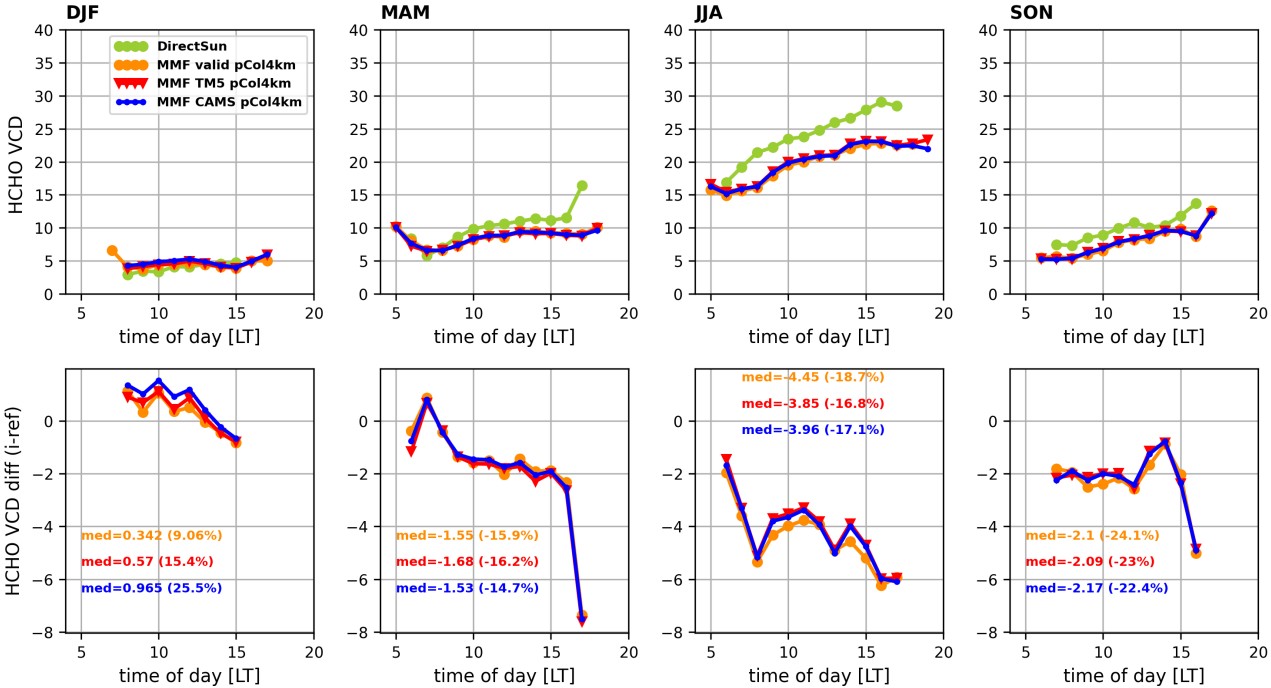

**Figure A13.** As Fig. 11 but for the MAX-DAOS pCol4km.

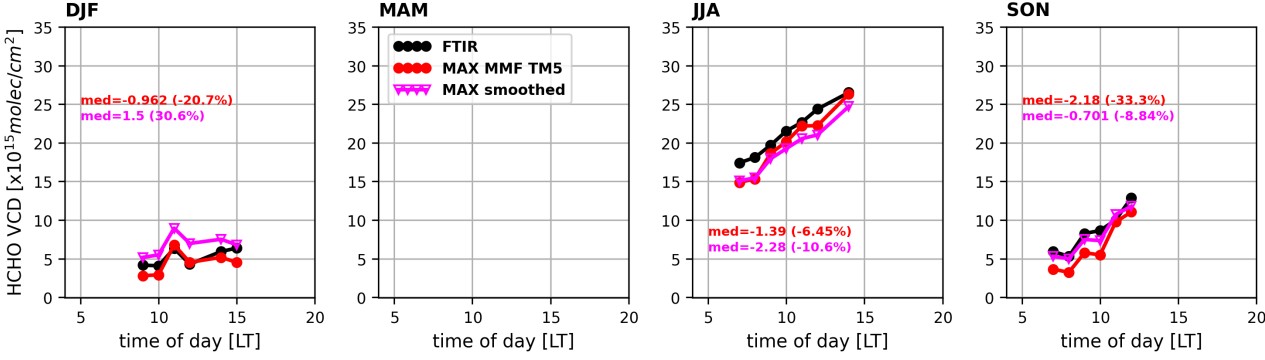

**Figure A14.** FTIR and MAX-DOAS (TM5 a priori test case) HCHO VCD diurnal variations per seasons. Both the original MAX-DOAS TM5 (red) and the substituted and smoothed one (magenta) are shown. Median statistics of the MAX-DOAS minus FTIR absolute and relative differences (M-F)/F are given for each season in the corresponding color. A minimum number of 10 points per hourly bin is required.




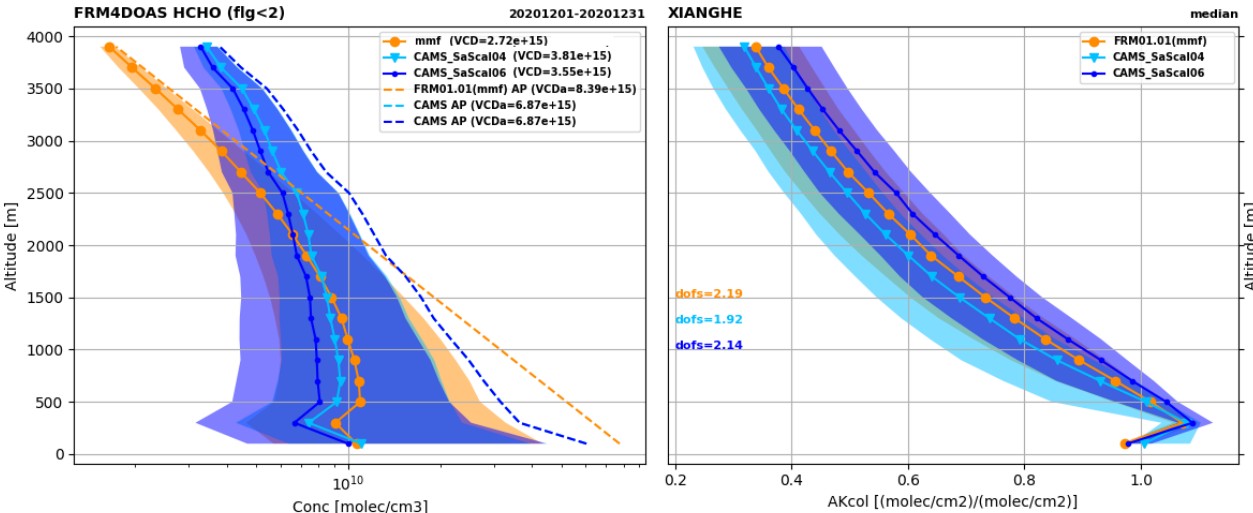

**Figure A15.** MAX-DOAS (MMF in orange), CAMS a priori test case for SaScal=0.4 (in cyan) and SaScal=0.6 (in blue) retrieved (plain lines) profiles (left panel) and corresponding averaging kernels (right panel) for the month of December 2020. The median values are shown with the addition of a shaded area representing the 25 to 75 percentiles values. The altitude is given in meters above the surface of the center of the retrieval layers.



*Author contributions.* GP carried out the investigations and wrote the manuscript. MVR, MMF, CV and BL contributed input and advise at all stages of the scientific discussions and of the manuscript writing. MMF, GP and CF prepared the ground-based MAX-DOAS FRM4DOAS data and MMF tests, MVR the direct sun, CV and BL the FTIR data. SB and TW are responsible of the MAPA data. CH, CV, TW and MZ installed the ground-based instruments and or supervised the instruments operation. IDS prepared the model data around the site. All co-authors revised and commented on the manuscript.

*Competing interests.* At least one of the (co-)authors is a member of the editorial board of Atmospheric Measurement Techniques.

*Acknowledgements.* Part of the reported work was carried out in the framework of the ESA FRM4DOAS-2.0 project, the ESA ATM-MPC TROPOMI validation activities and of the EUMETSAT AC SAF Continuous Development and Operations Phase (CDOP-4), and by the Belgian Federal Science Policy Office (BELSPO) via the ProDEx B-ACSAF contribution to the AC SAF. The FTIR HCHO measurements at Xianghe are supported by the China's national key research and development program (2023YFC3705202) and the State Key Laboratory of Atmospheric Environment end Extreme Meteorology (NO. 2024QN04). We would like to thank Nicolas Theys and François Hendrick for their support and advises at different stages of the study.



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
