# Peer review of "Intercomparison of MAX-DOAS, FTIR and direct sun HCHO vertical columns at Xianghe, China"

_EGUsphere, 2025_

## Author Response (AR1)

We thank the two reviewers for their constructive comments and suggestions that helped improving the paper. We answer the two reviewers in detail below, in blue. We also want to point out some major changes we made in the paper during the review.

1) While investigating the winter under-estimation after the smoothing step, we realized that we made a non-optimal choice for the extension of the MAX-DOAS substituted profile when regridding it on the FTIR altitude grid in Sect. 3.2.1. Actually, after the substitution step (eq 5 in the paper), we need to transform back this MAX profile on the FTIR grid, to be able to apply the FTIR AK (eq 6). For this step we used a regridding that was extending on the FTIR grid by aligning to the FTIR prior, above the MAX altitude height. But actually the FTIR prior is constant, and in winter the FTIR prior is much larger than the retrieved FTIR (see Fig A5a, now included as Fig 7 in the main part of the paper), this was strongly enhancing the MAXsubstituted_regridded_onF. In summer, it was the other way around, leading to a seasonal bias in the smoothed comparisons. Now, if instead of extending on the FTIR prior we extend on the retrieved FTIR profile, the seasonal bias disappears, so we updated the whole Sect. 3.2.1 and 3.2.2 smoothing part (Figures 8, 9, 12 and Table 4).

2) We revised Fig 11 where we compare the different MAX-DOAS MMF data to DS UV. The figure was originally done with all the available points but the strong impact of data sampling (especially in winter) lead to large differences in absolute and relative differences compared to Fig. 4. We decided to redo Fig 11(and A13, now A5), now only focusing on periods when all the datasets have valid data in a 15min interval.

3) Several figures in the appendix that were primarily diagnostic or redundant have been removed or merged. We removed A1, A2, A5, A6, A7, A8, A11, A12, A15 and simplified A4. We also moved to the appendix part of the discussions in Sect. 3.2. We hope these changes improve the readability and the visual density of the manuscript.

**Reviewer 1 (accepted subject to minor revisions)**

The authors present a valuable intercomparison study of formaldehyde (HCHO) vertical columns measured by three distinct ground-based remote sensing techniques. This work addresses a pertinent need for harmonized validation datasets, especially with the increase in satellite HCHO observations. The study is well-structured, and the methodological descriptions are clear. However, the current presentation reads more like a technical report than a scientific paper. Enhancing the readability by framing the study within a clear scientific question and discussing the implications of the findings would significantly strengthen the impact of this work. Given the technical soundness and the importance of the topic, I recommend Minor Revision after the following points are addressed.

We thank the reviewed for the presentation suggestion. We revised the abstract, introduction and conclusion keeping in mind the following scientific question "Are the HCHO retrievals consistent within the networks currently used for satellite validation (MAX-DOAS, FTIR, DS PGN)? Can we improve the MAX-DOAS?"

The abstract contains excessive methodological detail at the expense of clearly stating the core scientific findings and their implications. The introduction currently provides a good literature review but lacks a clear statement of the specific scientific gap this study aims to fill.

We simplified the abstract by removing many methodological details and revised the introduction stating more clearly the gap we want to fill.

The manuscript mentions that MMF and MAPA results often differ, leading to limited consolidated data output. However, it does not provide a clear, accessible explanation of why they differ, which is crucial for the community's understanding. In Section 2.1.1 and 2.1.2, the authors may want to include a concise summary highlighting the fundamental differences between the two MAX-DOAS retrieval approaches (e.g., optimal estimation vs. parameterization, use of a priori information). Consider adding a short paragraph or a table (in addition to Table 2) that contrasts the core philosophies, strengths, and inherent limitations of the MMF and MAPA algorithms. This will help readers understand the root causes of the observed discrepancies in HCHO VCDs.

We thank the reviewed and decided to add an introductory paragraph on in lines  .

The results show a significant underestimation (~22%) by MAX-DOAS compared to direct-sun measurements, primarily attributed to the a priori profile choice. The authors may need to expand the discussion to explore why the CTM-based profiles improve the agreement in some seasons but not in winter. The season-dependent behavior may deserve more deep discussion. Link it to the actual seasonal variation in the atmospheric HCHO profile structure over the site. Explain why the standard exponential decay profile is particularly inadequate in certain conditions.

As discussed above (see point 1), we revised the smoothing during the review, and we don't have such a season-dependent behaviour for the smoothed comparisons anymore. Moreover, we updated Fig 11 (see point 2) and this removed the wrong impression that using the model profiles was worsening the comparison in winter. However, winter is still a more sensitive period, with its small HCHO columns, the larger SZA (always above 60°, leading to larger errors for all the ground-based data, cf Fig A3), the smaller number of measurements, the increased sensitivity to data sampling and the larger differences between the two models. We added a discussion before discussing Fig 12, around lines 705-725 and in the conclusions (importance of winter).

The manuscript states the importance of the work for satellite validation but stops short of concrete recommendations. In the conclusion or a dedicated subsection, the authors may need to synthesize the findings into clear, actionable advice for the validation community.

We revised the conclusions adding some recommendations and discussions for the satellite validation.

**Citation**: https://doi.org/10.5194/egusphere-2025-3320-RC2

**Reviewer 2**

**RC1**: 'Comment on egusphere-2025-3320', Anonymous Referee #2, 26 Sep 2025 reply

**Summary of the manuscript**

This study presents an intercomparison of formaldehyde (HCHO) vertical columns retrieved from FTIR, MAX-DOAS, and direct sun (DS) DOAS measurements at the Xianghe station in China. The MAX-DOAS, and direct sun (DS) are measured using the same instrument. The work aims to assess the consistency of these ground-based techniques and to some extent evaluate the performance of the current FRM4DOAS MAX-DOAS retrieval system. The main goals of the study are: (1) assess the quality of the MAX-DOAS HCHO products currently delivered by the FRM4DOAS system and (2) revisit the HCHO retrieval approach used in the system to further improve its accuracy.

The authors first compare UV and IR DS HCHO VCD, finding good agreement. Comparisons between MAX-DOAS and DS/FTIR show that MAX-DOAS tends to underestimate HCHO by about 20%. They mentioned that when differences in vertical sensitivity and a priori profiles are accounted for, the bias is substantially reduced and further when the comparison to the 0-4 km range, where MAX-DOAS is most sensitive, also improves agreement. The study also investigates the impact of using chemistry transport model (CAMS and TM5) profiles as a priori in MAX-DOAS retrievals. Results show that this approach improves agreement in most seasons by better representing free-tropospheric HCHO contributions, although wintertime comparisons degrade slightly. Overall, the work demonstrates that MAX-DOAS HCHO underestimations primarily arise from limited sensitivity above 4 km and the choice of a priori profiles. The findings highlight the importance of harmonizing retrieval strategies across ground-based networks.

**Major Comments**

The manuscript provides valuable intercomparisons, especially because they incorporate the UV and IR Direct sun observations. However, the analysis is limited to a single site (Xianghe). The authors should discuss representativeness of suburban/urban conditions and how conclusions may differ in other regions, e.g., remote low HCHO levels and/or highly polluted regions. I do not recommend expanding the analysis to more sites but mentioning/reference how additional collocated sites (NDACC/FRM4DOAS) would strengthen the broader applicability of the findings.

We thank the reviewer for this comment. As mentioned in lines 549-554, there are other sites where a FRM4DOAS MAX-DOAS and an FTIR measure in parallel and we had performed preliminary comparisons at Bremen, Lauder, Toronto and Ny-Alesund, also finding MMF data being smaller than the FTIR VCD (see Figures ans1 to ans4 below for Bremen, Lauder, Ny-Alesund and Toronto). We commented this further in the conclusions, also stating that in some of these sites, a Pandora/PGN is also measuring HCHO (in direct-sun and off-axis geometry) and that further comparisons are needed.

[Figure]

Figure ans1: as Fig3 of the paper, but for FTIR vs MAX-DOAS MMF for Bremen over 2019-2022.

[Figure]

Figure ans2: as Fig3 of the paper, but for FTIR vs MAX-DOAS MMF for Lauder over 2019-2023.

[Figure]

Figure ans3: as Fig3 of the paper, but for FTIR vs MAX-DOAS MMF for Ny-Alesund over 2018-2023.

[Figure]

Figure ans4: as Fig3 of the paper, but for Toronto FTIR vs MAX-DOAS MMF for Bremen over 2019-2022.

Also, the authors should more explicitly highlight how this study advances harmonization and validation efforts. For example, my understanding is that MMF does not consider a correction factor for O4, is that correct?, while MAPA does. Have you seen the effect of using the same aerosol extinction profiles in both retrievals? Right now it is hard to see if aerosol is the cause of some of the discrepancies, what would authors recommend for FRM4DOAS given that MMF and MAPA results differ in too many cases as mentioned in the manuscript? An overall conclusion, suggestions, and guidance for improving the FRM4DOASr MAX-DOAS products is missing.

As a default run in FRM4DOAS, MMF is indeed not using any correction factor for O4, while for MAPA three versions are run in parallel, one with a scaling factor SF=0.8, one with SF=1 and one where the SF is a fitted parameter, which is the version considered in the paper. We compared the different MAPA results, and we made an additional run of MMF forcing a multiplicative factor in the O4 measured scd equal to the MAPA fitted SF. We show the results in Fig ans5, in the same style as the updated Fig 11 (see point 2) above). Results with the MAPA SF=0.8 are close to MAPA VAR results (the fitted SF is close to 0.8 in all the seasons (Fig ans6)) while results with SF=1 are strongly reducing the number of valid points, changing the statistics, especially in winter (and autumn). Results with the MMF variable SF are close to the original MMF ones, with some

differences in winter (and minor in autumn). The impact of the different data selection (sampling) in winter (and autumn) is clear.

[Figure]

Figure ans5: As updated figure 11, showing the DS, MMF and MAPA data for common coincident selection of points (the 3 datasets need to have valid data within the same 15 min bin). The different rows show different MAPA or MMF version: a) MAPA VAR (the default used in the study), b) MAPA with SF=1, c) MAPA with SF=0.8 and d) MMF forced with the fitted MAPA VAR SF.

[Figure]

Figure ans6: seasonal diurnal variation of the fitted O4 SF of the MAPA VAR retrieval.

Moreover, there are recent improvements of MMF that are being tested offline in FRM4DOAS to improve the AOD estimation, that consist in using the MAPA VAR fitted O4 SF in the MMF aerosols retrievals, showing more consistent AOD results wrt CIMEL (in the UV). We have made an additional MMF test using this approach and forcing a multiplicative factor to the O4 measured scd equal to the MAPA fitted SF, to be used within the HCHO retrieval, see Fig ans5d), dark green lines. We found that this is slightly increasing the winter and autumn HCHO VCD compared to the normal MMF (orange), but that this is not solving the seen differences wrt MAPA.

We included suggestions, and guidance for improving the FRM4DOAS MAX-DOAS products in the conclusions.

A central result is that the MAX-DOAS underestimation occurs from the choice of a priori, but the discussion of using CAMS/TM5-based priors remains somewhat unclear. The authors should clarify: how much of the improvement reflects a real reduction in bias versus an artificial correction; whether reliance on CTM priors risks transferring model biases; and the extent to which these results can be generalized to other sites.

We added some quantitative discussion in sect.3.2.2 about Fig A13 (now Fig A5), around line 700: "**To summarize, the impact of changing the exponentially decreasing a priori to model profiles, is thus leading to a ghost contribution added for the free tropospheric content in all the seasons of about 10%. In winter, when the HCHO content itself is small, there can also be some changes below 4 km, depending on the selected model.**"
We also added a more general discussion on model assessment in the conclusion, see answer to the question below.

Furthermore, both models used here are low spatial resolution (> 80 km), which will not capture heterogeneity from local sources. The inclusion of a high resolution model, in particular if using a single site for case study would benefit the paper, or a thorough information about the limitations of this is warranted.

We agree with the referee that having high resolution models could be interesting for better constraining the surface HCHO values, but to our knowledge there are no such model datasets runs freely available and implementing high-resolution regional GEOS-CHEM or WRF-CHEM models would deserve a specific study. Moreover, in principle we don't expect high spatial variability of HCHO columns in this suburban, where in principle anthropogenic sources are significant but only responsible for 7% of the total source (Stavrakou et al., 2009). As we show in the paper, using the models' profiles as a priori brings an added value where the MAX-DOAS is not sensitive, adding a free tropospheric HCHO content above 4km and both models are relatively similar in this regard (fig. 6, A4, A8), except for the winter CAMS less peaked profile feature. We added

some discussion about limitations of the coarse models, in lines 708-725, also referring to some recent studies focusing on HCHO profiles.

About the real reduction in bias versus an artificial correction and the reliance on CTM priors, this is indeed a drawback of introduction some dependences in the MAX-DOAS retrieval, but this is commonly done (FTIR and satellites rely on a priori, and HCHO satellite retrievals often need models to calculate a background correction, ie De Smedt et al. 2017). If, in addition of the retrieved profiles and the averaging kernels, we also provide the used a priori in the MAX-DOAS files, defined over the whole simulation grid (see Table 2 for the difference with the retrieved grid), the added ghost column information coming from these a priori can be inferred by the user. We have discussed this point further in the revised conclusion.

The degradation of agreement in winter when CTM priors are applied is mentioned but not fully explained. The authors should provide a stronger discussion of possible causes (e.g., reduced MAX-DOAS sensitivity, low HCHO levels, model representation issues rather than treating this as a minor limitation.

As discussed in point 1) at the beginning of the document and here below in detail, during the revision we found an inconsistency in the way the smoothing was done, that was leading to the degradation of agreement in winter. This is now fixed. We added anyway a discussion about the difference for CAMS in winter.

When harmonization (Rodgers & Connor approach) is applied, mean biases improve but regression slopes/intercepts sometimes degrade. This needs clearer explanation to avoid confusion about whether retrievals are actually improved and whether the smoothing process is actually needed, especially given the complexity of all the transformation variables.

As discussed above, we have revised the smoothing step and we realized that our current choice of regridding options (to bring the substituted MAXDOAS profile on the FTIR grid) was introducing a seasonality. We were using the FTIR prior above 4km, but as the FTIR prior is fixed, it was introducing some biases by leading to larger values than the FTIR retrieved in winter and smaller values in summer (see previous Fig A5 now in Fig 6). We clarified this step in the manuscript, adding some explanations between eq 5 and eq 6, and we revised the choice to use the FTIR retrieved profile itself for the extension above 4km. This solved the worse regression results after the harmonization. We also extended the MAXDOAS to FTIR comparisons to the whole 2018-2021 period, resulting in this new Fig 8 and new Fig9:

[Figure]

The substitution and smoothing are reducing the bias in all the seasons now. In summer and winter, the smoothed MAX VCD is now slightly larger than the FTIR itself, but all the differences are now around ±6%, which is the median difference for the FTIR to DS UV comparisons.

We also added a line with the 2018-2021 results in Table 4 that was previously only presenting the values for 2020, so that the numbers can be compared to those when using CAMS and TM5, that have only been processed for 2020. We updated all the numbers in Table 4 and added the variability on the median relative bias as half the interpecentile68.

We agree with the reviewer that the smoothing process can be complex and that many possibilities on how to perform it exist and can impact the results. But we think it is important doing the smoothing to understand well the MAX-DOAS to FTIR comparisons and we recommend doing it when exploring the instrumental differences in detail. Moreover, for satellite validation, considering that FTIR and satellite have similar AK compared to the MAX-DOAS ones, it is important to use them, when available.

Several figures (especially in the Appendix) are dense and difficult to interpret. Simplification is warranted. The manuscript is already quite long, and the large number of complex figures makes it challenging to follow. I recommend retaining only those figures that clearly support the main findings and improve readability.

We have simplified the Appendix by removing figures A1, A2, A5, A6, A7, A8, A11, A12, A15 and simplifying A4 and moving to the appendix part of the pCol4km and H75 discussion from the main part of the paper.

Since one of the main motivations is satellite validation (e.g., TROPOMI, GEMS, TEMPO, etc), the manuscript should discuss more concretely how improved MAX-DOAS retrievals will influence satellite bias assessments and network harmonization. This would enhance the relevance and impact of the study.

We have included some discussion about the impact of the MAX-DOAS harmonization for the validation in the conclusions.

The comparison between FTIR and direct sun DOAS is limited to a few months, even with a gap, within a single year. I recommend to include in the conclusions that long-term comparisons are still warranted to test stability over long-term.

We agree that longer UV vs IR direct sun comparisons would be beneficial. We mentioned this in the conclusion and added a recommendation to compare more systematically HCHO DS data of FTIR, Pandora PGN, and MAX-DOAS data where possible.

**Specific comments**

P1, L10: The median difference is reported as a negative number, but without further explanation this sign is not meaningful. I recommend a short explanation and also include uncertainty/variability among all quantitative results.

We added explicitly that the difference is for (FTIR-DS)/DS and added the variability (estimated as half the interpercentile68) on it: -6% ±10%

P1, L12: For readers unfamiliar with FRM4DOAS, the reference is unclear. Please define FRM4DOAS or use a more general description.

We removed the acronym and extended a bit the description.

P1, L15: When discussing the underestimation (-22% and -20.8%), please include the associated variability. Are these values significantly different, or can they be summarized as a ~20% underestimation?

Thanks for the remark. We added the variability estimation as half the interpercentile68 next to the biases and discussed it in the main parts of the paper (text and tables). It is in general between 17 and 35% and the differences with respect to FTIR and DS UV can

indeed be summarized as an underestimation of ~20%. We kept it simple in the abstract.

Abstract: The distinction between the two bias reductions (to 1% after accounting for a priori/AKs vs. 2.5% after restricting to 0–4 km) is not clear. These paragraphs appear redundant or inconsistent. Please clarify the difference in approach and interpretation.

The two bias reductions are related to 1) considering the whole atmosphere in the MAX-DOAS to FTIR comparisons (and extending in a way or another the MAX-DOAS profile above it maximum retrieval altitude of 4km, see main introduction above) and 2) considering in the comparisons only the MAX-DOAS vertical retrieval range, and cutting the comparisons at 4km, ie similar to comparing pCol4km, see tests in Table 4. We agree that in the current stage the description was not clear enough, and we decided to remove the explanation of this result in the abstract, to simplify it, as suggested by the other reviewer.

Abstract / Introduction: It is stated that MAX-DOAS has no sensitivity above 4 km, yet the underestimation is attributed to the a priori. This is contradictory. If improved agreement comes only from using better a priori profiles, then the improvement is not due to additional information from MAX-DOAS but rather the imposed prior. Please clarify.

MAX-DOAS MMF start from the a priori, and it deviate from it only where it has sufficient sensitivity. Moreover, in FRM4DOAS the retrieval is only performed up to 4km, so if the prior is not zero above this value, the integrated retrieved profile and the total simulated VCD can differ, by a value defined by the prior content above 4km, a "ghost column". In this sense, the change of prior (from an exponentially decreasing profile to a model profile including some free tropospheric content above 4km) lead to a larger MAX-DOAS VCD, and thus a reduction of the under-estimation. But indeed, this is not an additional information from MAX-DOAS but a ghost column addition, related to the choice of the used prior.

We changed the sentence in the abstract to "**The underestimation in the current MAX-DOAS VCDs is thus coming from the limited vertical sensitivity of the technique and from the choice of the a priori profile, which neglects the free-tropospheric contribution (above 4km), where the MAX-DOAS has no sensitivity. We test and suggest possible improvements to the current centralized MAX-DOAS HCHO retrievals processing, like using more appropriate a priori profiles, based on the CAMS and TM5 chemical-transport models (CTMs) that better estimate the HCHO content above 4km.**"

P2, L26: Define what is meant by "positive impact" when describing the use of CTM-based priors.

As discussed above and following the changes in results obtained for the revised smoothing part, we decided to simplify the abstract and remove these methodological description, and keep it simple: " **We test and suggest possible improvements to the current centralized MAX-DOAS HCHO retrievals processing, like using more appropriate a priori profiles, based on the CAMS and TM5 chemical-transport models (CTMs) that better estimate the HCHO content above 4km.**"

Text (paraphrased, P2):

"When restricting the comparison to the 0–4 km altitude range, the impact of the a priori profile is only significant in the winter period, also leading to a degradation of the agreement with FTIR data. The improvement of the agreement between MAX-DOAS and FTIR data is thus mainly related to a better handling of the free-tropospheric part of the profile, smaller in winter than in other seasons."

This sentence is unclear. Does it mean that the comparison worsens in winter and improves during the rest of the year? Please rephrase to make the seasonal dependence explicit.

Yes, this was the meaning of the sentence. However following the changes in results obtained for the revised smoothing part and in the interest of simplification, we removed this paragraph completely from the abstract.

P4, L102, Why do you mean by "state-of-the art", I recommend removing that.

Done.

P4, L106: Include the altitude of the Xianghe station.

Xianghe station altitude is 26m asl. Added.

P5, L132: Please explain why the UV channel stopped operating in 2018. Was there a technical failure or another reason? Overall, the sentence "The Xianghe MAX-DOAS measurements cover the period…. for the UV channel. The VIS channel continued to work until August 2022" is not clear because before it is mentioned that "It is a dual channel system composed of two grating spectrometers covering the UV and visible". What happened after 2022?, is the instrument currently working and be used for future satellite validation?

In 2018 the sun-tracking system had a failure and needed to be repaired. It was back on operations in October 2019, allowing both DS and MAXDOAS geometries. The instrument is a dual channel, in the sense that it includes two spectrometers (one for the UV and one for the VIS) that are synchronized, but that can also operate autonomously. In end of 2021, the UV detector start to be unstable and then broke, and no good UV measurements are thus possible from Nov 2021, while the VIS part of the

instrument continued working nominally up to mid 2022, when the VIS CCD also started having problems. Since then, the instrument is out of operations.

We rephrased sentence 132-133 from "The Xianghe MAX-DOAS measurements cover the period from March 2010 to July 2018 and from October 2019 to November 2021 for the UV channel. The VIS channel continued to work until August 2022." to "**The Xianghe MAX-DOAS measurements cover the period from March 2010 to November 2021 for the UV channel and up to August 2022 for the VIS channel. Between July 2018 and October 2019 there is a data gap due to a failure of the pointing system. The instrument is not operating anymore**."

P5, L147: Spell out acronyms MMF and MAPA at first mention.

done

P8 (a priori column): Please justify why 8.4 x 1015 molec/cm2 is used as the a priori vertical column value.

This value is the default value used in FRM4DOAS for HCHO for all the sites (except Arctic and Antarctic). It can be seen by looking to Fig A9 that this value is within the Xianghe VCD yearly values. Anyway, in MAX-DOAS the a priori VCD value is not so important, the shape of the a priori profile is more important.

P8, L205: Correct "1rst" to "1st."

Done

Figure 2: How is the direct-sun averaging kernel in panel (b) estimated? Is it the total column AK? If so, please state this explicitly, and consider also including the FTIR total column AK for comparison.

The total column averaging kernels for UV direct-sun geometry (and corresponding airmass factors) were estimated using a ray-tracing model accounting for earth sphericity and the temperature dependence of the HCHO absorption cross sections. We use the formulation developed by Eskes and Boersma (2003) for optically thin absorbers, which relates the averaging kernel to the airmass factor calculation. The HCHO a-priori profile is based on monthly averaged CAMS model simulations for the month of June. Tests using a range of other profiles covering a full year show negligeable dependency (<1%, see Fig ans7 below). In Figure 2b) it is indeed a total AK that is shown (please note we have added more curves up to 85°SZA in panel b)). We added a sentence in the figure caption. The total AK comparison for DS, FTIR and MAX-DOAS is presented in Fig7, so we decided not to add it in Fig2 to keep it simpler.

[Figure]

Figure ans7: HCHO DS UV AKs difference with respect to the annual mean as a function of SZA, for different months of the year.

Section 2.3: If the FTIR instrument is mainly operated for TCCON-like observations, what is the effective time resolution of HCHO retrievals?

We had typically 1 point every hour (see line 400) when there is sun (and after solving the measurement issues described in lines 408.

Section 2.4: Given the coarse spatial resolution of the models, please describe in more detail how a priori profiles are extracted at the Xianghe site. A clearer explanation of the interpolation method would be helpful.

The model profiles (TM5: 1°x1°, 30 min, CAMS REA: 80kmx80km, 3h) have been considered as they are used in the L2 and L3 S5p data for the CCI+precursor project. First the model profiles are included in the L2 S5p files, adapting them for each pixel. Then L3 are created from these L2, and we used the cell covering Xianghe. Details are given below and We included some additions on this in the section, lines 383.

- In the L2 algorithm, for each satellite ground pixel, the profiles are linearly interpolated in space and time, at pixel centre and S5P local overpass time. To reduce the errors associated to topography and the lower spatial resolution of the model compared to the satellite spatial resolution, the a priori profiles need

to be rescaled to effective surface elevation of the satellite pixel. Following Zhou et al. (2009) and Boersma et al (2011), the model surface pressure is converted by applying the hypsometric equation and the assumption that temperature changes linearly with height (ref: TROPOMI HCHO ATBD, De Smedt et al., 2018). The pressure levels for the a priori HCHO profiles are based on the improved surface pressure level (with p=a+b*ps and a, b the constants that effectively define the vertical coordinate of the model).

- For the L3, the profiles in vmr (and other quantities such as the surface pressure) have been averaged on a regular grid (resolution: 0.125°) on a monthly basis using the HARP atmospheric toolbox (ex: harp_product_bin_spatial in https://stcorp.github.io/harp/doc/html/libharp_product.html).

- The grid cell corresponding to Xianghe is selected at the end of these steps.

Section 3.1.1: It appears averaging kernels are not applied in the FTIR vs MAX-DOAS DS comparison. Please confirm if this is correct, and if so, justify why it is not necessary to account for the AK differences.

Indeed, DS AK are not applied to the FTIR profile in Fig 3, Sect 3.1.1 as the DS UV AK are close to 1 compared to FTIR ones, see Fig7 and see the revised Fig2b) where we have added a few more curves for the DS AK for more intermediate SZA values. Compared to the AK of the other instruments, the DS UV AK are only deviating significantly from 1 for SZA around 75° and above (increasing the HCHO profile below 2km height and decreasing the profiles above), and we don't have many such conditions in the FTIR measurements. We have made a test for a few days applying the DS UV AK profile for SZA=75 or 85° to FTIR profile, with changes on FTIR VCD smaller than 0.5%. We added this info in lines 416-422.

[Figure]

[Figure]

[Figure]

FTIR (2020-01-01 04:32:33, 62.9°SZA), VCD=7.06e+15, VCDsmooth=7.07e+15
FTIR (2020-01-01 05:50:19, 66.6°SZA), VCD=7.2e+15, VCDsmooth=7.23e+15
FTIR (2020-01-01 07:07:57, 74.6°SZA), VCD=7.55e+15, VCDsmooth=7.57e+15
change in percent: 100*(smooth-orig)/orig:
[0.15937458 0.46045259 0.34784981]
FTIR (2020-06-30 22:40:58, 70.9°SZA), VCD=2.41e+16, VCDsmooth=2.42e+16
FTIR (2020-07-01 05:02:46, 19.4°SZA), VCD=3.38e+16, VCDsmooth=3.4e+16
change in percent: 100*(smooth-orig)/orig:
[0.07089272 0.42281359]
FTIR (2020-12-15 01:34:45, 72.4°SZA), VCD=3.92e+15, VCDsmooth=3.94e+15
FTIR (2020-12-15 02:45:46, 65.8°SZA), VCD=4.01e+15, VCDsmooth=4.02e+15
change in percent: 100*(smooth-orig)/orig:
[0.39206137 0.2637398 ]

Section 3.1.1 (reference spectrum): Please describe when and how the MAX-DOAS DS reference spectrum was derived. How sensitive are the results to the season/time of year chosen for the reference?

The direct-sun reference spectrum was selected on June 15, 2020 at local noon. The residual slant column amount of HCHO in this spectrum ($2.7\pm0.16 \times 10^{16}$ molec/cm$^2$) was evaluated from a full year of measurements using the Bootstrap estimation method (Cede et al., 2006). This uncertainty estimation is confirmed by repeating the same procedure using different reference spectra. We added these details in the text.

The differences with a DS UV dataset using different reference spectra is of the order of 6% (see lines 273-278).

Figure 3: Was any filtering applied to the quantitative correlation analysis? Please clarify.

As indicated in line 361, we only consider FTIR data within +/-30 min of the DS and interpolated on a common temporal grid. So, no, no averaging for the correlation analysis, but interpolation of one data set on the other (excusing points too far away).

P7, L378: The direct-sun DOAS (DS) is chosen as the main comparison reference for MAX-DOAS in Sect. 3.1.2, although FTIR offers a longer overlap period. Please elaborate on the reasoning for prioritizing DS over FTIR in this section.

As a first step we use DS as a reference for all the datasets (FTIR and MAX-DOAS) because the DS UV provide more frequent sampling than the FTIR, closer to the MAX-DOAS sampling (during 2020, 552 points for FTIR and more than 5000 for DS, about the same number for MAPA valid data, and about 10000 for MMF valid data). So, for the VCD comparison, we first rely on DS reference, but then we also compare to FTIR where the additional step of comparing the profiles can be explored we extended the comparisons of Sect. 3.2.1 to the 2018-2021 period (updated Fig 8 and 9), but kept the smoothing comparison to 1 year only when discussing the use of models as a priori in Table 4 and Fig 12 and A6.

P12, L271: It is mentioned that a fixed reference spectrum is used for the entire DS period, but also that using a season-specific reference spectrum reduces the fit residual while increasing the uncertainty. This seems counterintuitive, could you explain why this occurs?

Using regularly updated reference spectra (e.g. one per season) improves the fitting residuals due to the reduced impact of small drifts in the instrumental spectral response. Nevertheless, a fixed reference spectrum was preferred to avoid possible jumps in the time-series related to the uncertainty in the determination of the residual HCHO SCD in each reference spectrum.

P13, L311. It is mentioned that FTIR AK peaks around 10 km and is about 0.8 at the surface, what would it mean a value of 2 around 10km?

A value of 2 around 10km means larger contribution of the a priori profile in the retrieval, which is too sensitive to the variability at this height. But at this height the HCHO concentration is quite small (see red dotted curve in Fig 7), so this should not impact too much the whole retrieval.

P14, L335: Improve caption of Figure A4. What is the difference between each subpanel using the same model?

The lower panels are a zoom from 0 to 10km of the first row (left is CAMS, right is TM5). We decided to remove the second row with the zooms, to simplify a bit the figures in the annex.

P17, L374: I suggest removing Lines 374-375 as this does not contribute to the findings ans the paper is already long.

We removed Fig A6 and removed the sentence but decided to keep the 5 coincident points as the default choice for the whole manuscript figures instead of 10, to still have some comparisons in spring with FTIR.

P17, L378: It is interesting that authors have chosen UV DS for the comparison with MAX-DOAS because the FTIR covers a longer time span, please clarify why this was chosen.

As answered to comment "P7, L378" above, we actually use both UV DS and FTIR as reference (sect. 3.1.1, 3.1.2 and then 3.2.1), but we start from the UV DS one. We extended to the 2018-2021 period for Sect. 3.2.1.

P17, L403: What reasoning or analysis is carried out to mention that Pandora direct sun product overestimates HCHO? Or how did you derive this conclusion?. It is mentioned that "difference found in the latter study is larger than what a free tropospheric HCHO could explain" but not explanation.

This is coming from the study of Fu et al., 2025 on GEMS comparisons. We adapted the sentence (lines 467) to "**However, the authors of the latter study claim that the difference found... and they suggested that, in some sites, ... . One illustration is e.g. the comparisons in Bae et al. (2025)...**"

P19, L418: Please clarify why partial columns up to 4 km were specifically chosen.

4km is the highest altitude of the MMF profile retrieval grid, so we used this altitude to separate the purely retrieved profile contribution from the information coming from the whole simulation input. Above 4 km, if the a priori is not zero, it will be considered within the simulation, and its content will be added as a ghost column in addition to the retrieved profile, to the total VCD. 4km is also used for the additional comparison to the FTIR columns.

P19, L431: Define H75 when first introduced.

We checked the first occurrence and adapted accordingly.

Section 3.2. Profile comparisons. I recommend revising this section thoroughly for clarity. At the moment, it is very difficult to follow, especially with the large number of figures in the appendix. It would be better to streamline the text and focus on the most important points.

We moved the discussion on pCol4km and H75 in the appendix (and revised and reduced the figures in the appendix to simplify them).

Figure 5. The figure shows profiles from MAX-DOAS (MMF and MAPA) up to 4km. I am a bit confused, does the retrieval of MAX-DOAS for both MMF and MAPA are carried up only up to 4 km?. Typically, retrievals should be carried up with more layers and when no information is coming from the observations a priori information would be used.

The FRM4DOAS output is providing profiles only up to 4km, as for MMF the retrieval is indeed only performed up to 4 km, and above the a-priori information is used. In the default MMF case, the a priori is zero above 4km by construction, so VCD and pCol4km

agree, while when using the models as a priori, VCD is larger than pCol4km. For MAPA, the retrieval is performed up to 20km but the output profile is only reported on the same output grid than MMF, up to 4km, see table 2 (simulation, retrieval and output grid information). For MAPA, there is no a priori.
This is a change we suggest for FRM4DOAS in the conclusions, see around line 800.

P23, L508: It appears that the median bias decreases with smoothing; however, as noted, the overall correlation worsens. I recommend including the uncertainty or error (e.g., standard deviation) associated with the bias, since the variability also seems to increase with smoothing. In addition, Figure 9 indicates that smoothing has a negative effect in winter. Based on these points, please discuss whether smoothing is still necessary or justified for future studies.

As discussed above, we have revised a step of the smoothing procedure and now the regression results are better than the original ones. We think smoothing is necessary to take into account the difference in vertical sensitivity and the impact of different priors. Using pCol4km (or similar) could be an option if there is enough information, but we lose the information on the impact of the different priors.

Figure 7: How was the DS averaging kernel derived? Please describe the method in the text.

As discussed above, we added the following discussion in Sect. 2.2:

**"The total column averaging kernels for UV direct-sun geometry (and corresponding airmass factors) were estimated using a ray-tracing model accounting for earth sphericity and the temperature dependence of the HCHO absorption cross sections. We use the formulation developed by Eskes and Boersma (2003) for optically thin absorbers, which relates the averaging kernel to the airmass factor calculation. The HCHO a-priori profile is based on monthly averaged CAMS model simulations for the month of June. Tests using a range of other profiles covering a full year show negligeable dependency (<1%). "**

Figure 8. What is the difference between bias and (med) in the text for each subplot?

Bias is the mean, med is the median. We revised the figure to clarify this.

Table 4 caption: Clarify what "original" vs. "smoothed" means. Does "original" mean without smoothing, and "smoothed" mean extended to 100 km and then convolved with FTIR AKs? Please make it explicit.

Yes, original is without smoothing and smoothed is actually with the full Rodgers and Connor approach of eq 6 (profile substitution + application of FTIR AK). We added this in the table.

P9, L509 / Figure 8: It is stated that smoothing improves the bias, but this is not convincingly shown in Fig. 8b. While the median bias appears to improve, the regression slope degrades, and at high columns MAX-DOAS underestimation is worse while at low columns an overestimation appears. This suggests the improved bias is an artifact of offsetting errors rather than a genuine improvement. The seasonal plot in Fig. 9 (DJF) makes this issue more evident. Please provide an explanation and discuss whether smoothing is still necessary or appropriate for future intercomparisons.

As discussed for comment "P23, L508" and above, we have revised the way the smoothing was done, fixing now the seasonality issue. See our comment "P23, L508" for the importance of the smoothing.

Section 3.1.1 (sensitivity): Please clarify the importance (or lack thereof) of differences in sensitivity between direct sun IR and UV retrievals. This may be relevant to interpreting the results.

As discussed above, the DS UV AK are only deviating significantly from 1 for SZA around 75° and above (see Fig2b), and we don't have many such conditions in the FTIR measurements. We added the following text in lines 416-420:

"In this comparisons we only focused on the VCD and we did not took into account the difference in sensitivity of the DS UV and FTIR, as this is small. The DS UV AK are only deviating significantly from 1 for SZA values around 75° and above (with larger values up to 1.04 below 2km for the extreme 85°SZA case), while the FTIR AK are a bit smaller than 1 (with values down to about 0.8) below 1 to 1.5km and larger above, see Fig. 2b and Fig. 7. Using the largest DS AK (for SZA=85°) on the FTIR profiles would typically increase the HCHO profile below 2km height and decreasing the profiles above, but we don't have many FTIR measurements for SZA>75°. We made a test for a few days applying the DS UV 85°SZA AK to FTIR profile, with changes on FTIR VCD smaller than 0.5%."

P30, L650: Please clarify what the reported "–6%" refers to. As written, it is not clear whether FTIR or DS DOAS values are lower. Specify explicitly which dataset underestimates the other.

Ok, we reformulated. -6% is (F-D)/D (see Table 3), so on average, FTIR smaller than DS UV

References

Eskes, H. J., & Boersma, K. F., Averaging kernels for DOAS total-column satellite retrievals. *Atmospheric Chemistry and Physics*, 1285–1291, 2003.

Cede, A., Herman, J., Richter, A., Krotkov, N., and Burrows, J.: Measurements of nitrogen dioxide total column amounts using a Brewer double spectrophotometer in direct Sun mode, Journal of Geophysical Research: Atmospheres, 111, https://doi.org/10.1029/2005JD006585, 2006.

Zhao, T., Mao, J., Zhao, X., Pandey, A., Shah, V., Knowland, K. E., et al. (2025). Summertime diurnal variability of formaldehyde over the contiguous United States: Constraints from pandonia global network. *Geophysical Research Letters*, 52, e2025GL116033. https://doi.org/10.1029/2025GL116033

De Smedt, I., Theys, N., Yu, H., Danckaert, T., Lerot, C., Compernolle, S., Van Roozendael, M., Richter, A., Hilboll, A., Peters, E., Pedergnana, M., Loyola, D., Beirle, S., Wagner, T., Eskes, H., van Geffen, J., Boersma, K. F., and Veefkind, P.: Algorithm theoretical baseline for formaldehyde retrievals from S5P TROPOMI and from the QA4ECV project, Atmos. Meas. Tech., 11, 2395–2426, https://doi.org/10.5194/amt-11-2395-2018, 2018.

Zhou, Y., Brunner, D., Boersma, K. F., Dirksen, R., and Wang, P.: An improved tropospheric NO2 retrieval for OMI observations in the vicinity of mountainous terrain, Atmos. Meas. Tech., 2, 401–416, https://doi.org/10.5194/amt-2-401-2009, 2009.

Boersma, F. K., Eskes, H. J., Dirksen, R. J., van der A, R. J., Veefkind, J. P., Stammes, P., Huijnen, V., Kleipool, Q. L., Sneep, M., Claas, J., Leitão, J., Richter, A., Zhou, Y., Brunner, D., and Veefkind, P.: An improved tropospheric NO2 column retrieval algorithm for the Ozone Monitoring Instrument, Atmos. Meas. Tech., 4, 2329–2388, https://doi.org/10.5194/amt-4-1905-2011, 2011.

---

## Author Response (AR2)

We thank the editor for his revision and technical correction suggestions. We adapted the text accordingly.

Dear authors,

Thank you for your revision and responses to reviewers' comments. I have now reviewed them and noted some examples of technical corrections (please see below) needed as some of them may not be fully captured during the final editing process. I recommend publication subject to minor revisions.

There are certain acronyms that are not defined in their first use (e.g., ESA, AP, VCD, etc.). Please identify and define them. VCD -> vertical column density.

Page 1, line 11: '-0.5 10" -> "-0.5 x 10".

--> done

Page 7, line 154: Missing end of sentence after "correlations".

--> done

Page 7, line 166: "produce" -> "produces".

--> done

Page 8, line 177: "does not work as well as for" -> "does not work well as for" .

--> done

Page 8, line 178: "increase" -> "increases".

--> done

Page 8, Eqn 2: Please check Eqn. 2 for the absolute sign.

--> done

Page 15, line 385: "did not took" -> "did not take".

--> done

Page 17, Table 3: please define IP68/2 either in the table or the text.

--> done

Page 19, line 422: I think "present" isn't necessary.

--> done

Page 19, line 432: Please remove "e.g."

--> done

Page 19, line 435: "Bae" -> "Bae et al."

--> done

Page 19, line 442: For clarity, it may help to add "for the year" before "2020" here and in other places to give a context of a year.

--> done

Page 21, line 475: "fix" -> "fixed"

--> done

Page 25, line 535 and page 26, line 554: What do you mean by raw difference? I suggest removing raw if there is no specific meaning in these cases.

--> done

Page 27, Figure 10: "apriori" - "a priori" here and in several other places.

--> done

Page 34, line 748: "E.g.," -> "For example,"

--> done

Page 34, line 751: "in Pinardi et al. (2013) Figure 18" -> "in Figure 18 of Pinardi et al. (2013)".

--> done

Page 37, line 778: "H75 is increase" -> "H75 increases"

--> done

Sincerely,

Lok Lamsal, PhD

AMT Associate Editor